# Low-Rank Tensor-Network Encodings for Video-to-Action Behavioral Cloning

**Brian Chen**[*]                                                                 *chenbri@umich.edu*
*Department of Mathematics*
*University of Michigan*

**Doruk Aksoy**[*]                                                                 *doruk@umich.edu*
*Department of Aerospace Engineering*
*University of Michigan*

**David J. Gorsich**                                                      *david.j.gorsich.civ@army.mil*
*Ground Vehicle Systems Center*
*U.S. Army*

**Shravan Veerapaneni**                                                          *shravan@umich.edu*
*Department of Mathematics*
*University of Michigan*

**Alex A. Gorodetsky**                                                            *goroda@umich.edu*
*Department of Aerospace Engineering*
*University of Michigan*

**Reviewed on OpenReview:** *https://openreview.net/forum?id=w4DXLzBPPw*

## Abstract

We describe a tensor-network latent-space encoding approach for increasing the scalability of behavioral cloning of a video game player's actions entirely from video streams of the gameplay. Specifically, we address challenges associated with the high computational requirements of traditional deep-learning based encoders such as convolutional variational autoencoders that prohibit their use in widely available hardware or for large scale data. Our approach uses tensor networks instead of deep variational autoencoders for this purpose, and it yields significant speedups with no loss of accuracy. Empirical results on ATARI games demonstrate that our approach leads to a speedup in the time it takes to encode data and train a predictor using the encodings (between $2.6\times$ to $9.6\times$ compared to autoencoders or variational autoencoders). Furthermore, the tensor train encoding can be efficiently trained on CPU to achieve comparable or better training times than the autoencoder and variational autoencoder trained on GPU ($0.9\times$ to $5.4\times$ faster). These results suggest significant possibilities in mitigating the need for cost and time-intensive hardware for training deep-learning architectures for behavioral cloning.

## 1 Introduction

Learning by imitation is a paradigm that enables teaching complex tasks through observation of expert behavior (Hussein et al., 2017) including object manipulation (Shen et al., 2022), robotic locomotion (Wang

---

[*]: Equal contribution
DISTRIBUTION A. Approved for public release; distribution unlimited. OPSEC#7367

et al., 2022), and autonomous driving (Bronstein et al., 2022). Several approaches for learning from data have been proposed, including behavioral cloning and inverse reinforcement learning.

In this work, we consider the task of learning directly from expert demonstrations in the form of action and video-frame sequences. In this context, learning directly from raw pixel data is computationally challenging due to the high resolution of game frames. This challenge is often addressed via downsampling and usage of large-scale computational resources. An illustrative example is the OpenAI VPT (Baker et al., 2022) for the game of Minecraft, which first downsamples the original frame size of $640 \times 360$ to $128 \times 128$. Furthermore, the model is trained using $70,000$ hours of gameplay, has half a billion parameters, and is trained for nine days on $720$ V100 GPUs.

One approach to tackle the computational challenges of using raw video frames as input is to provide hand-tuned features (such as movement, location, terrain, visibility, etc.) rather than, or in addition to, raw pixel data. With this additional information, important elements of the game are provided and thus do not have to be learned solely from video data. This is the approach taken by some reinforcement learning and imitation learning works, such as Berner et al. (2019) for Dota 2 and Vinyals et al. (2019) for Starcraft II. However, such an approach requires human effort and insight in identifying traits that would be useful for a learning algorithm and developing a means for extracting this data. As an alternative to defining hand-crafted features, there have been various approaches proposed for learning a latent space representation via self-supervised learning, where auxiliary tasks are used to train a neural network that maps high-dimensional input data into a lower dimensional space (Chen et al., 2021b). In this setting, autoencoder-based deep learning architectures have been a common method of choice for identifying this latent space (Jaques et al., 2021; Brown et al., 2020; Chen et al., 2021a;b; Amini et al., 2018), although other approaches have also been explored (Sermanet et al., 2017; Pari et al., 2021; Chen et al., 2021b).

In this work, we explore latent space extraction via a more computationally efficient tensor network approach, based on the Tensor Train (TT) format (Oseledets, 2011). The TT-format offers a multi-linear approximation that can be tuned to have a target reconstruction accuracy without the need of extensive hyperparameter tuning. Furthermore, the recent development of an incremental algorithm, `TT-ICE`* (Aksoy et al., 2024), allows us to compute TT-decomposition of a large amount of video data faster and with less memory than the standard `TT-SVD` algorithm — a critical enabling innovation for this application. `TT-ICE`* computes the TT-decomposition entirely using basic linear algebra operations such as matrix multiplication and decomposition. Furthermore, it only requires a single pass through the data and provides compression guarantees, a significant advantage over backpropagation techniques that can require multiple passes through the training data and do not guarantee any representation accuracies. The lack of guarantees on representation accuracy by deep-learning architectures also leads to the need for validation-data driven hyperparameter tuning to find hyperparameters and architecture settings that lead to high-quality representations. Instead, the complexity of the TT representation, represented by the tensor *rank*, can be found adaptively to guarantee a target accuracy on the training data.

While the TT-format may be viewed as having a smaller representational capacity compared to the nonlinear architecture of neural networks, there is evidence suggesting that neural networks might exhibit more expressiveness than required for specific tasks. Several works have found that some layers of trained neural networks can be replaced with tensor networks with little to no degradation of accuracy while improving the computational and memory costs of running these networks (Novikov et al., 2015; Garipov et al., 2016; Yu et al., 2017b; Kim et al., 2016; Lebedev et al., 2015; Sharma et al., 2021). In our work, we test whether encodings provided by neural networks can similarly be replaced by tensor networks. However, in contrast to the above works, we train the tensor network encodings directly, instead of first training a neural network encoding then replacing it (or components of it) with tensor networks.

In this paper, we demonstrate that this approach does provide a practical and highly effective alternative to deep-learning based encodings by reducing the requirements of hyperparameter tuning, thereby offering shorter training times for behavioral cloning. Moreover, the TT-encoding-based predictors achieve similar or higher behavioral cloning accuracies than predictors built on encodings from a convolutional autoencoder (AE) or variational autoencoder (VAE) (Kingma & Welling, 2014). Although other neural network ap-

proaches for feature extraction are available (see Chen et al. (2021b) for examples), we chose AEs and VAEs since like the TT, these are trained to minimize a loss that is related to reconstruction error.

To summarize, our contributions are as follows:

1. Empirical demonstration that the low dimensional embeddings obtained through `TT-ICE`* can be used for the downstream task of behavioral cloning.

2. Empirical demonstration of a reduction in training time ($2.6 - 9.6\times$ faster when trained on GPU), while achieving comparable behavioral cloning accuracy and gameplay scores compared to autoencoder-based approaches for extracting a latent space.

## 2 Related Work

In machine learning literature, AEs and VAEs are commonly used as feature extraction tools. Polic et al. (2019) use a convolutional autoencoder (CAE) to extract features from images of a tactile sensor. The extracted features are then used to estimate contact object shape, edge position, orientation and indentation depth using shallow neural networks. Yu & Zhou (2020) extract features from gearbox vibration data using CAE and uses the extracted features to identify fault modes. Pintelas et al. (2021) use CAE to compress and denoise high dimensional images to a smaller size and then use the encoded images for classification, Zabalza et al. (2016) use a stacked AE to extract features from hyperspectral images and use those features in classification, and Cai et al. (2022) use AE-extracted-features to determine the level of traffic congestion from video frames.

There is also increased attention to transformer based methods. Yang et al. (2022) use a transformer encoder to generate pair-wise relations for the objects in the image and use a transformer decoder to generate captions. Kim et al. (2021) use a transformer based architecture as a feature extraction step in dual arm robot manipulation. However, training complex architectures like transformers increases the computational cost and data requirements even further. In Kim et al. (2021), it is reported that 8 NVIDIA V100 GPUs were used in training.

In addition to neural network-based tools, tensor methods have also been considered as a feature extraction option in machine learning literature. Tensor methods allow the extraction of meaningful features using linear algebra operations, therefore eliminating the need for backpropagation. Taherisadr et al. (2019) use CANDECOMP/PARAFAC (CP) format to reduce the size of EEG signals and increase the computational efficiency of a CNN-based seizure predictor. Zhang et al. (2017) use Tucker format to extract features from PCG signals and a support vector machine based classifier to identify anomalous heart sounds. Fonał & Zdunek (2019) use TT-format to extract features from various multidimensional datasets and a $k$-nearest neighbor classifier to perform multi-class-classification. Zhao et al. (2020) use a TT layer to extract features from high-dimensional video data before passing those features into a recurrent neural network for the task of video summarization.

Although tensor methods and neural networks are both capable of feature extraction, the question of whether the nonlinearity of neural networks can lead to features that are better suited for downstream tasks than the multilinear tensor methods remains. Previous studies indicate that the increased expressiveness of neural networks might be excessive for certain tasks and computationally expensive compared to similar tensor-based methods. Several works have explored compressing portions of large neural networks after training, including fully connected layers (Novikov et al., 2015) and convolutional layers (Garipov et al., 2016; Yu et al., 2017b; Kim et al., 2016; Lebedev et al., 2015; Sharma et al., 2021). The fine-tuning of large pre-trained neural networks can also be simplified by exploiting low rank structure; Hu et al. (2022) explore using low-rank matrices to update the weights of large language models when fine-tuning these models for downstream tasks. Other work has also explored using tensor operations directly within the network architecture itself during training as a means for parameter reduction and model compression; examples include Tensor Train Recurrent Neural Networks (Yang et al., 2017), LR-TT-DNN (Qi et al., 2023), Tensor Contraction Layers (Kossaifi et al., 2017), Tensor Regression Layers (Kossaifi et al., 2020), and Higher-Order Tensor RNNs (Yu et al., 2017a). The ability to replace portions of deep neural networks with

tensor networks either during or after training suggests that alternative representations may exist that do not require as much computational power and time to train, while not sacrificing accuracy. However, such works propose embedding tensor networks *within* neural networks and therefore require optimization and fine-tuning to capture the nonlinearity of the compressed layer(s) accurately. As such, they don't mitigate the issues surrounding the complexity of optimization approaches.

In this work, we explore using a tensor network in TT-format as an alternative to neural network based approaches, where the TT is used to extract features from video frame data for behavioral cloning. This approach is newly made possible by incremental algorithms (such as TT-ICE (Aksoy et al., 2024)) with approximation guarantees, which provide the basis for memory efficient training of these models (in contrast to TT-SVD of Oseledets (2011), or other optimization-based incremental approaches). In comparison to using neural network based encodings like convolutional AEs and VAEs, we show that using a TT leads to significantly faster training times without sacrificing behavioral cloning accuracy.

## 3 Methodology

The overarching objective of this work is to extract a *game-specific* playing strategy from video demonstrations of an experienced agent. In this section we first describe our learning framework; then we present the methods to extract a low-dimensional representation and discuss how TT-decomposition fits within this setting. Finally, we briefly present the action predictor that maps latent representations to actions.

The imitation learning framework we use is shown in Figure 1. It consists of two parts: an encoding transformation (labeled "Frame encoder" in Figure 1) and an action predictor ("Action predictor" in Figure 1). The encoding transformation serves to reduce the dimensionality of the space so that the action predictor can potentially be learned more efficiently. In this paper we aim to accelerate computation of the encoding transformation.

The encoder works by mapping frames of a video sequence into an $L$ dimensional latent-space. We compare three such encoders in this work: two neural network-based encoders determined from an AE and VAE, and a low-rank encoder based on the TT-decomposition. The dimensionality of the latent space is typically determined by the formulation of the encoding transformation. For example, AEs and VAEs would treat this as a hyperparameter over which to optimize, whereas our low-rank approach will adaptively determine this dimension to a user-specified error tolerance. We further note that each of the three methods outlined above also have an ability to reconstruct an image from the latent space, which potentially can be useful for investigating the latent space.

Although there are other neural-network approaches for learning a latent reprsentation, these mostly change the loss / auxiliary objective that is used to train the network. For instance, whereas AEs/VAEs are trained to minimize reconstruction error, contrastive methods instead learn to minimize a distance between related samples (such as two distorted views of the same image (Chen et al., 2020)) and maximize a distance between unrelated samples. Because the main difference is the training objective, the training time for AEs/VAEs should be similar to contrastive methods, assuming similar neural network architectures. We benchmark our results of using the TT against AEs/VAEs due to the similarity in their training objectives, which is to minimize reconstruction error.

In the rest of this section, we will discuss the methods and tools to compute the latent space representation of the gameplay sequences efficiently in detail.

### 3.1 Frame Encoder

This section discusses three alternatives for frame encoding. We will first briefly introduce AEs and VAEs and then discuss our TT-based approach in detail.

As mentioned in Section 1, AEs and VAEs have found great use for extracting encodings in the context of imitation learning and behavioral cloning. However, low-rank tensor decompositions have not been widely used for this purpose.

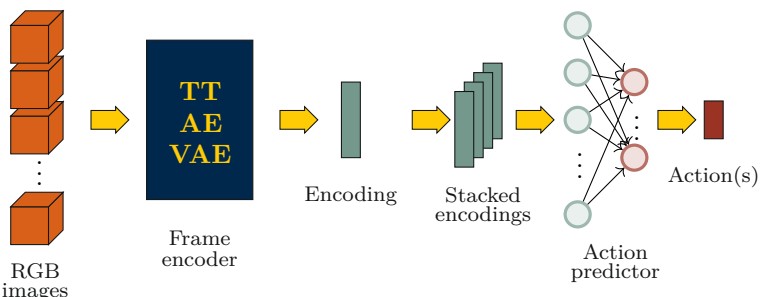

Figure 1: *Imitation learning framework. Our focus is to enhance latent space learning depicted by the blue box. We compare the TT, autoencoder (AE) and variational autoencoder (VAE) approaches. Orange cubes represent images from gameplay videos. The extracted latent features are then used in the action-prediction network to predict actions. The architecture of the action predictor remains the same for all three encoding methods.*

### 3.1.1 Autoencoder

AEs are a neural network based latent space extraction tool. AEs are used for a variety of tasks, including dimensionality reduction (Wang et al., 2016), denoising (Majumdar, 2018), and image compression (Sreelakshmi & Ravi, 2020). AEs consist of two parts: an encoder and a decoder. The encoder takes the input data and maps it to a latent space representation. The decoder then takes the latent space representation and maps it back to the original data. The goal of the AE is to learn a mapping that minimizes the difference between the original data and the reconstructed data.

AEs are a powerful tool for unsupervised learning, and they can be used to extract meaningful features from data without any prior knowledge. However, AEs can be computationally expensive to train, and they can be sensitive to the choice of hyperparameters.

### 3.1.2 Variational Autoencoder

VAEs (Kingma & Welling, 2014) are similar to AEs in utilizing an encoding and decoding network. However, instead of mapping a fixed input to a fixed latent vector, the input is mapped probabilistically to a latent vector $z$. A benefit of VAEs is that it is a generative model, allowing one to generate new samples by sampling latent vectors $z$ (typically) from a standard normal distribution and passing it through the decoder. An additional potential benefit is that the training of a VAE imposes some structure onto the latent space, as the latent space is trained to be normally distributed.

The AE and VAE are both implemented using an encoder consisting of four convolutional layers, followed by a flattening layer and a final linear layer, and a decoder consisting of a linear layer, a reshaping layer, and four transposed convolutional layers. Additional details on the architecture and training of the AEs and VAEs can be found in Appendix A.

### 3.1.3 Tensor-Train (TT) Format

The TT is a compact representation method for multi-dimensional arrays. A $d$-dimensional tensor $\mathcal{X} \in \mathbb{R}^{n_1 \times n_2 \times \cdots \times n_d}$ is said to be in *TT-format* if it is represented as a product of $d$ 3-dimensional tensors $\{\mathcal{G}_i\}_{i=1}^d$, $\mathcal{G}_i \in \mathbb{R}^{r_{i-1} \times n_i \times r_i}$, called *TT-cores*. The internal dimensions $r_i$ of TT-cores $\mathcal{G}_i$ are also referred to as *TT-ranks* and are determined by error-truncated SVDs through a user-specified relative error tolerance $\varepsilon \in [0, 1]$, except for $r_0 = r_d = 1$.

In the case of RGB images, a video sequence is a 4-dimensional tensor $\mathcal{X} \in \mathbb{S}^{H \times W \times C \times N}$ with $\mathbb{S} = [0, 255]$. The first three dimensions correspond to the height (H), width (W), and color (C) channels; and the last dimension corresponds to the index of an individual image within the video sequence.

In the video application considered in this paper, we first reshape $\mathcal{X}$ into a 6-way tensor of shape $n_1 \times \ldots \times n_5 \times N$, where $\prod_{i=1}^{5} n_i = HWC$. By reshaping $\mathcal{X}$ to have smaller dimensions, we aim to limit the maximum TT-rank for each dimension. This also results in a slower growth in overall TT-ranks and yields higher compression. However, increasing the number of dimensions may result in slower computational speed because the TT-format is computed sequentially. In other words, increasing the number of dimensions will increase the number of TT-cores and therefore increase the number of SVDs needed to compute the TT-decomposition.

After reshaping, the tensor of a video sequence is represented in TT-format as

$$\mathcal{X} \approx \mathcal{G}_1 \ _3\times_1 \ \mathcal{G}_2 \ _3\times_1 \ \cdots \ _3\times_1 \ \mathcal{G}_6, \tag{1}$$

where $\mathcal{A} \ _3\times_1 \ \mathcal{B}$ denotes the tensor contraction between the third dimension of $\mathcal{A}$ and first dimension of $\mathcal{B}$, and $\mathcal{G}_i \in \mathbb{R}^{r_{i-1} \times n_i \times r_i}$. This representation often leads to a reduction in the size of the data. The original (but reshaped) data has a storage complexity of $O(n^d)$, where $n = \max_{i \in \{1,\ldots,d\}} n_i$. The TT-format has storage complexity $O(ndr^2)$, where $r = \max_{i \in \{0,\ldots,d\}} r_i$.

Note that in Equation 1, the first 5 TT-cores, $\mathcal{G}_1, \ldots, \mathcal{G}_5$, correspond to reshapings of the H, W, and C of the images – these are common to all images. The final dimension indexes the unique frame and therefore only the final core is unique to a data point. To be more specific, for the $i$-th image stored in $\mathcal{X}$, we have the reconstruction

$$\mathcal{X}(i) \approx \mathcal{G}_1 \ _3\times_1 \ \cdots \ _3\times_1 \ \mathcal{G}_5 \ _3\times_1 \ \mathcal{G}_6[:, i], \tag{2}$$

where $\mathcal{G}_6[:, i]$ denotes the $i$-th column of $\mathcal{G}_6$. As a result of this, we can use the representations stored in $\mathcal{G}_6$ as encodings of the frames and the fifth TT-rank, $r_5$, becomes the size of the latent space $L$.

One of the methods to compute a TT-decomposition for $d$-dimensional arrays is the `TT-SVD` algorithm (Oseledets, 2011, Alg. 1). This algorithm returns an approximation $\tilde{\mathcal{X}}$ that satisfies the inequality $\|\mathcal{X} - \tilde{\mathcal{X}}\|_F \leq \varepsilon_{des} \|\mathcal{X}\|_F$ for some predefined relative error upper bound $\varepsilon_{des}$. Moreover, when computed with `TT-SVD`, the first 5 TT-cores consist of reshapings of orthonormal vectors (i.e., left singular vectors). However, performing the `TT-SVD` on large tensors requires loading the full tensor into memory and performing a sequence of singular value decompositions. As a result, this approach requires a significant amount of memory. The memory of our existing computational resources limits us to a maximum number of around 30,000 frames for `TT-SVD`.

One way to reduce the memory and computational requirements of `TT-SVD` is adopting an incremental approach. `TT-ICE`* (Aksoy et al., 2024, Alg. 3.2) is an incremental algorithm that computes the TT-decomposition for a stream of tensors with a given target accuracy. To train the tensor network, we create batches of $B = 2500$ frames from gameplay sequences and update the basis vectors stored in TT-cores using `TT-ICE`*. Although `TT-ICE`* does not require a fixed batch size as shown in Aksoy et al. (2024), we train the tensor network using a fixed batch size to mimic the training of the NN based methods. Once a pass is completed over a batch of images, their latent representations are stacked to $\mathcal{G}_6$. Note that while `TT-SVD` computes the TT decomposition of a $n_1 \times \ldots \times n_d \times N$ tensor in time $O(dn^{d+1}N)$ (Huber et al., 2017), `TT-ICE`* computes a tensor train using batches over the final dimension. Letting $B$ be the batch size and $b$ be the total number of updates, `TT-ICE`* calculates the TT decomposition in time $O(bdn^{d+1}B)$. If all batches are used to update the TT, this leads to the same computational complexity as TT-SVD (with $bB = N$). However, since `TT-ICE`* learns a generalizable space, not all $B$ frames from a batch are needed to update the TT-cores. `TT-ICE`* updates the TT-cores by selecting the subset of frames that have an approximation error higher than the threshold. Furthermore, in the edge case where the TT-cores are highly representative of a batch, `TT-ICE`* skips updating TT-cores and directly computes the latent space embeddings using Algorithm 1. Through these modifications `TT-ICE`* achieves $bB \leq N$. Please refer to Table A.1. in Aksoy et al. (2024) for more details. Even for the worst case of $bB = N$, `TT-ICE`* offers a substantial reduction in memory needed to compute the TT-decomposition in addition to eliminating the necessity of storing all frames at once.

Also note that `TT-ICE`* explores and increases the number of the basis vectors for each dimension of the tensor stream adaptively based on the user-specified error tolerance. This removes the need to manually tune the size of the latent space $L$ for the encoded images.

In case a problem has some special properties to allow an *a-priori* upper bound on TT-rank, the rank and complexity must be adapted in response to received data. The datasets used in this work do not readily come with useful upper-bounds on the TT-ranks. As a result, we set a relative error tolerance of 1% for each training batch and the TT-ranks are automatically determined by `TT-ICE`* (please refer to Theorem 2 and Theorem 4 of Aksoy et al. (2024) for proofs of approximation error)[1]. In the asymptotic limit of the data, `TT-ICE`* will be able to represent any tensor because of its rank adaptation property.

Next, we discuss how out-of-sample gameplay frames are mapped to the latent space by each encoding method.

### 3.2 Mapping unseen data to latent space

For AE/VAE, the latent space representation of unseen data is acquired directly through a forward pass of the trained encoder with the desired image. On the other hand for TT, the notion of forward pass is replaced with sequential projection onto cores.

Note that the first 5 TT-cores contain reshaped orthonormal vectors when TT-cores are trained with `TT-ICE`*. We obtain $r_i$ orthonormal vectors for each dimension by reshaping the first 5 TT-cores as

$$U_i = \texttt{reshape}\left(\mathcal{G}_i, [r_{i-1}n_i, r_i]\right). \tag{3}$$

These orthonormal vectors are then used to encode any new images from the same game type. We obtain the latent embedding of images by simply projecting the appropriately reshaped images onto $U_i$. The pseudocode of this procedure is provided in Algorithm 1[2]. Note that as the ranks $r$ of the TT increase, the computational time of Algorithm 1 grows quadratically with $r$. In particular, the number of flops for the projection grows at a rate of $O(drn^{d-1}N\max(r,n))$, where $N$ is the number of new images to project, $n = \max_{i\in\{1,\dots,d\}} n_i$, and $d$ is the dimensionality of the tensor. For the application considered in this paper, $n = 16$ and $d = 5$.

The AE, VAE, and TT all allow for reconstructing frames from the latent space. As such, we can visualize the reconstructed frames to get a sense of what is lost in encoding the frames. Figure 2 shows an example of reconstructed frames from the latent space for each encoding method.

---

**Algorithm 1** Mapping unseen data to latent space by orthogonal projections onto TT-cores

---

1: **Input**
2:      $\{\mathcal{G}_i\}_{i=1}^5$          TT-cores trained with `TT-ICE`*
3:      $\mathcal{Y}_j \in \mathbb{R}^{n_1 \times \cdots \times n_5 \times N_j}$    $j$-th batch of frames that contain $N_j$ frames to be projected
4: **Output**
5:      $g_j \in \mathbb{R}^{r_5 \times N_j}$          encodings of the $j$-th batch of frames
6: $g_j \leftarrow \texttt{reshape}\left(\mathcal{Y}_j, [n_1, n_2 \ldots n_5 N_j]\right)$
7: **for** $i = 1$ to $4$ **do**
8:      $U \leftarrow \texttt{reshape}\left(\mathcal{G}_i, [r_{i-1}n_i, r_i]\right)$                      ▷ Note that for $i = 1$ we have $r_0 = 1$
9:      $g_j \leftarrow U^T g_j$
10:      $g_j \leftarrow \texttt{reshape}\left(g_j, [r_i n_{i+1}, n_{i+2} \ldots N_j]\right)$
11: **end for**
12: $U \leftarrow \texttt{reshape}\left(\mathcal{G}_5, [r_4 n_5, r_5]\right)$
13: $g_j \leftarrow U^T g_j$

---

### 3.3 Action predictor

To extract the game-specific playing strategy, we employ a simple MLP based architecture. We stack the latent representation of the last 4 frames together and obtain a $4L$ vector as input to the action predictor. Then the action predictor outputs the predicted action for a given timestep. Using the previous four frames as

---

[1]When the *occupancy* heuristic is not used, the approximation error guarantees proven for `TT-ICE` apply to `TT-ICE`* as well.
[2]This algorithm is adapted from Aksoy et al. (2024)

Figure 2: *Example reconstruction of a Ms. Pac-Man frame using TT (with $\varepsilon = 0.01$), AE and VAE. TT reconstruction is visually the most similar to the original frame. The size of the latent space for all three models is 11583, which is $11.5\%$ of the original size ($210 \times 160 \times 3$).*

input allows the model to use some amount of temporal information and is a common choice for experiments conducted on Atari games (see, e.g., Mnih et al. (2015)).

The action predictor consists of 5 hidden layers with 50 neurons each. Every hidden layer is then followed by a batch norm and ReLU operation. Finally, the output layer has size equal to the number of possible actions for each game, which is interpreted as the probability of taking each action. Next, we present the computational experiments we have conducted.

## 4 Experiments

In this section, we describe our comparative study, the dataset we selected for the experiments, evaluation metrics we used, and results we obtained.

### 4.1 Dataset

The dataset consists of game frames from demonstrations from a set of Atari games, in particular: Beam-Rider, Breakout, MsPacman, Pong, Qbert, Seaquest, and SpaceInvaders[3]. Our data is collected using trained Deep Q-Networks (Mnih et al., 2015), which are obtained from the RL-Baselines Zoo package (Raffin, 2018) and further trained using the Stable Baselines platform (Hill et al., 2018) for $10^7$ samples (states and actions), using the tuned hyperparameters from RL-Baselines Zoo. The chosen Atari games were selected due to the availability of trained models and tuned hyperparameters in the RL-Baselines Zoo package. The reinforcement learning algorithm used to generate the model (Deep Q-Networks) was chosen both due to the availability of trained models and tuned hyperparameters as well as the fact that Deep Q-Networks have been shown to perform well on Atari games (Mnih et al., 2015).

Rather than using downsized grayscale $84 \times 84 \times 1$ images (before stacking) as is often done in the literature, we instead use the original full-sized RGB images ($210 \times 160 \times 3$), since our work deals directly with image compression and dimensionality reduction. For each game, we generate 200 trajectories, a subset of which is used as a training set and a subset of which is used as a validation set. More details about the split are given in Section 4.2. Another 50 trajectories are generated as a test set, which is used to evaluate the action prediction accuracy of the final trained models. We define a trajectory as a sequence of states (frames), rewards, and actions taken by the RL agent, starting at the beginning of the game and ending when the RL agent loses a life in the game or the end of the game is reached. The dataset can be found at Chen (2023).

---

[3]We note that we have also tested on Enduro, but due to computational limitations for the AE/VAE, results are not presented. In the moderate data case, the AE / VAE took roughly 30 hours to train per seed compared to 4.9 hours for TT-GPU.

## 4.2   Setup

The procedure for our experiments is as follows:

1. Train an AE, VAE, and TT on the frames from a set of gameplay videos. The dimensionality of the latent space $L$ for the TT is determined automatically by `TT-ICE`$^*$ . After the TT is trained, an AE and VAE are trained with a latent space size fixed to be equal to the size $L$ of the latent space from the TT. This is done to provide a fair comparison of the methods. The latent size $L$ for the various games are shown in Table 1. Furthermore, the VAE and AE are trained using a range of L2 regularization coefficients and initial learning rates.

2. For the AE and VAE, an additional portion of the gameplay videos are held out for validation. The validation reconstruction errors are used to select a VAE and AE for encoding. Note that for TT-decomposition no such validation procedure is needed.

3. Using the encodings generated by the TT or the AE / VAE as input, we train simple feed-forward networks to predict the actions taken by the trained reinforcement learning agent from which data is generated, using a range of hyperparameters.

4. For each of the AE, VAE, and TT encodings, we select the predictor that yields the lowest validation action prediction loss, as measured by cross-entropy loss on a separate validation set.

In our work, we consider training the AE, VAE, and TT separately to first learn an encoding, before training a predictor on the encodings directly. This is in contrast to end-to-end approaches, where the encodings would be trained jointly with the predictors. An advantage of training the encoders separately from the action predictors is reduced training time, as the expensive encoders are not further finetuned for the particular task of behavioral cloning. A second advantage is that the encodings are smaller than the full frames, leading to lower memory and storage requirements. This approach mimics the approach taken in several reinforcement learning papers, which have found that decoupling training an encoder from training the reinforcement learning agent is an effective approach - examples include Pari et al. (2021); Parisi et al. (2022); Stooke et al. (2021); Yuan et al. (2022); Shah & Kumar (2021).

The predictors and AE / VAE were set up and trained using PyTorch and trained on NVIDIA Tesla V100 GPUs. The TT is trained either using GPU (TT-GPU, NVIDIA Tesla V100) or CPU (TT-CPU, Intel Xeon Gold 6154 processors), using the TT-ICE* package - timings are reported for both. Note that the only difference between TT-GPU and TT-CPU is the device that the computation took place. This is achieved by replacing NumPy functions (Harris et al., 2020) used for computations with their CuPy counterparts (Okuta et al., 2017) for TT-GPU. In parallel to the findings in Aksoy et al. (2024), we have empirically found that this approach is most effective when $n_i = \mathcal{O}(10)$ for $i = 1, \ldots, 5$ and therefore reshaped the image batches to $15 \times 14 \times 16 \times 10 \times 3 \times N$, where $N$ is the batch size used for the TT (in our experiments, $N = 2500$).

In this work, we consider two cases: 1) a limited data regime where only 5 gameplay trajectories are used as training data for each game and 2) a moderate data regime where 160 gameplay trajectories are used as training data for each game. A summary of the number of trajectories used in the limited and moderate data settings is shown in Table 2. In addition to these two scenarios, we also provide a more detailed investigation on the effects of changing the latent space size (Appendix B.2) and the amount of training data (Section 4.4.3) on action prediction accuracy and computational speedup.

Action predictors are trained with the Adam-W optimizer (Loshchilov & Hutter, 2019) with a cosine learning rate decay (Loshchilov & Hutter, 2017) for fifty epochs in the low data case and five epochs in the moderate data case. These predictors are trained using the same range of initial learning rates and weight decay penalties as the ones used to train the AE and VAE encoders (shown in Appendix A). An additional 1 or 20 trajectories (in the limited and moderate data regimes, respectively) not used during training are then used as a validation set to evaluate the trained predictors, with the predictors yielding the lowest cross-entropy action prediction loss selected.

Table 1: *Latent sizes L for the AE, VAE, and TT for the games tested. Note that the latent size is determined automatically for the TT given the prescribed relative error bound of* 0.01. *That latent size is then used for the AE and VAE.*

|  | Latent Size $L$ |
|---|---|
| BeamRider | 14645 |
| Breakout | 3561 |
| MsPacman | 11962 |
| Pong | 1287 |
| Qbert | 2267 |
| Seaquest | 15163 |
| SpaceInvaders | 17191 |

Table 2: *Summary of the number of trajectories used to train the models in the limited and moderate data cases. Note that each row corresponds to separate trajectories.*

|  | Limited Data | Moderate Data |
|---|---|---|
| Training Set for Training Encoder and Predictor | 5 | 160 |
| Validation Set for Selecting Best Encoder (ONLY AE/VAE) | 1 | 20 |
| Validation Set for Selecting Best Predictor | 1 | 20 |
| Test Set for Evaluating Final Models | 50 | 50 |

We also include the results of a predictor trained using the original, full-sized RGB frames, of dimension $210 \times 160 \times 3$. Because the "no encoding" predictor uses the full-sized images, it is expected that these accuracies will be higher than the predictors using the encodings - these are included as a comparison as an upper bound on accuracy we can expect.

Additional details on training and the neural network architectures can be found in Appendix A.

### 4.3 Metrics

We compare performance of the methods using four metrics: training time, reconstruction quality, action prediction accuracy, and normalized gameplay score. We then report the averaged results for each of those metrics for each regime in Section 4.4.

**Training Time.** We compare the time required to train each framework against the time needed to train a predictor on full frames. The training time for each framework can further be broken down into the time it takes to train the encoder, the time it takes to encode the data, and the time it takes to train the action predictor given the encodings. These results are shown in Appendix B.1 as Figures 9 and 10.

**Reconstruction Quality.** We measure the reconstruction quality of the TT against an AE and VAE using peak-signal-to-noise ratio (PSNR) and mean-squared error (MSE) as metrics.

**Action prediction accuracy.** We compare the accuracy of the actions suggested by each framework to actions taken by the RL agent. The output of the behavioral cloning agents is a vector of probabilities for each action - we calculate the proportion of times the action with the highest probability is equal to the action with the highest probability outputted by the RL agent.

**Normalized gameplay score.** We normalize gameplay scores achieved by each framework so that a score of 0 corresponds to performance of an agent taking random actions and a score of 1 corresponds to performance of the RL agent. In particular, given the score $s$ achieved by a behavioral cloned agent, the mean score $s_{\mathrm{RL}}$ achieved by the RL agent and the mean score $s_{\mathrm{random}}$ achieved by an agent taking random

actions, we calculate the normalized score to be

$$s_{\text{normalized}} = \frac{s - s_{\text{random}}}{s_{\text{RL}} - s_{\text{random}}}.$$

### 4.4 Results

#### 4.4.1 Limited Data Regime

The benefits of bypassing the need for training a neural network encoder can be observed in the total training time of the different methods, as shown in Figure 3. In particular, the training time of the TT-based predictor trained on GPU (TT-GPU) is 8.2× to 9.6× faster compared to the AE and VAE-based predictors and 26.1× to 29.7× faster compared to the predictor trained with no encoding. When the TT encoding is trained on CPU (but with the predictor still trained on GPU), the TT-predictor is still 3.3× to 5.4× faster than the AE and VAE predictors trained on GPU and 10.4× to 17.5× faster than the predictor trained with no encoding.

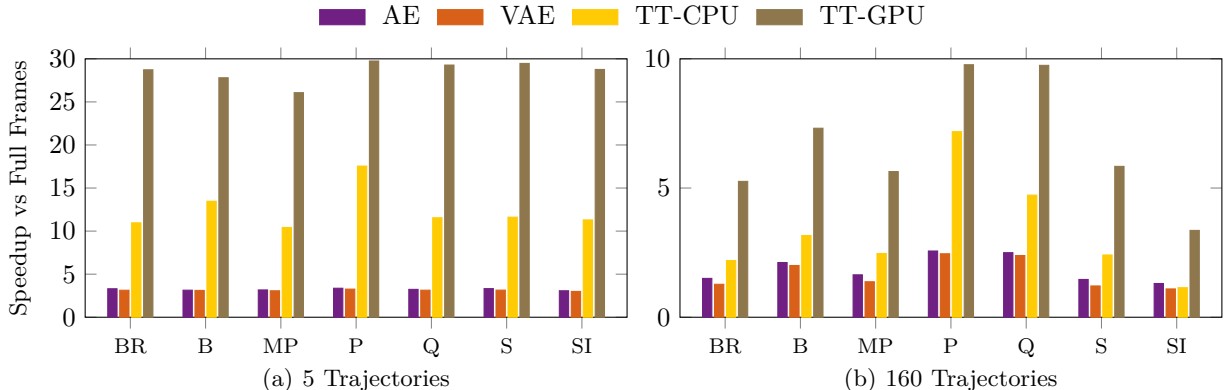

Figure 3: *Speedup of the end-to-end training procedure of the AE, VAE, and TT-based predictors compared to a predictor trained using the full RGB frames. Extracting a latent space reduces end-to-end training time in all cases. In the low data regime, using the TT-GPU leads to a 26.1× to 29.7× faster training time than using the full RGB frames and a 8.2× to 9.6× speedup compared to the AE and VAE. In the moderate data case (160 trajectories), using TT-GPU leads to a 3.4× to 9.8× faster training time compared to using the full RGB frames and a 2.6× to 4.8× speedup compared to the AE and VAE. Note that the difference in speedups between the 5 trajectory case and 160 trajectory case is due to the difference in training epochs used in each case (50 in the 5 trajectory case and 5 in the 160 trajectory case); the speedups are similar when the number of training epochs is held fixed while the number of training trajectories is changed, as shown in Section 4.4.3. BR:Beam Rider, B:Breakout, MP:Ms.Pacman, P:Pong, Q:Qbert, S:Seaquest, SI:Space Invaders.*

This speedup comes with an improvement in reconstruction quality, as shown in Table 3; the TT leads to a lower MSE and higher PSNR compared to the AE and VAE in six of the seven games. This speedup also comes at no cost in terms of behavioral cloning accuracy (Figure 4a) and achieved gameplay scores (Figure 5a), compared to training using encodings generated by an AE or VAE.

#### 4.4.2 Moderate Data Regime

In the case where 160 trajectories are used for training data, we find once again that the TT-based predictor provides a large speedup relative to the AE, VAE, and full frame predictors. In particular, as shown in Figure 3, using the TT-encoder trained on GPU led to a 3.4× to 9.8× faster training time compared to the no-encoding predictor. Compared to the AE and VAE, the end-to-end training time of the TT-based predictor is 2.6× to 4.8× faster. Training the TT on CPU still usually leads to a speedup over the AE and VAE on GPU (0.9× to 2.9× faster), showing the computational efficiency of TT-ICE even on CPU compared to GPU-based deep learning.

These results come with an improvement in reconstruction quality and no cost in terms of behavioral cloning accuracy or gameplay scores. The reconstructions generated by the TT have lower error compared to

Table 3: *Reconstruction quality of the TT, AE, and VAE in the low data regime. The reported statistics are MSE (using 255 as maximum pixel value) and PSNR of the reconstructions versus the original frames. For each game, the bolded values are the smallest MSE and highest PSNR.*

| | MSE | | | PSNR | | |
|---|---|---|---|---|---|---|
| | TT | AE | VAE | TT | AE | VAE |
| BeamRider (BR) | **6.3** | 30.5 | 47.6 | **46.8** | 35.3 | 33.0 |
| Breakout (B) | **33.0** | 67.6 | 80.9 | **39.4** | 32.9 | 30.3 |
| MsPacman (MP) | **27.1** | 58.0 | 103.3 | **38.7** | 32.5 | 28.9 |
| Pong (P) | 2.0 | **0.7** | 56.9 | 46.7 | **61.6** | 36.9 |
| Qbert (Q) | **6.8** | 33.5 | 68.5 | **45.6** | 34.3 | 30.8 |
| Seaquest (S) | **13.9** | 30.8 | 74.0 | **38.6** | 33.9 | 29.8 |
| SpaceInvaders (SI) | **80.3** | 144.5 | 176.4 | **32.4** | 28.4 | 26.2 |

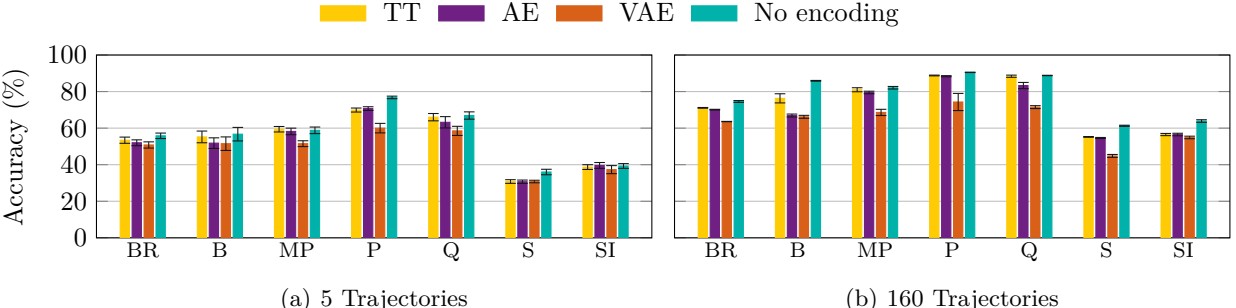

(a) 5 Trajectories        (b) 160 Trajectories

Figure 4: *Action prediction accuracies of trained predictors using encodings from the TT, AE, and VAE, as well as a trained predictor using the full-sized original RGB frames. Training a predictor using TT encodings yields similar and sometimes better prediction accuracies as training a predictor using AE or VAE encodings. Error bounds are calculated as two times the standard error of gameplay scores calculated across ten (in the 5 trajectory case) or three (in the 160 trajectory case) experiments. Please refer to the caption of Figure 3 for the game abbreviations.*

the reconstructions generated by the AE and VAE, as shown in Table 4. Furthermore, we find improved behavioral cloning accuracy of the predictor trained using the TT encodings versus the predictors trained on the AE or VAE encodings for some of the games, as seen in Figure 4b. The gameplay scores achieved by each of the encoding methods is comparable, as shown in Figure 5b.

Note that in some games, the gameplay scores achieved by the predictors, including the one trained with no encoding, are much lower than the scores achieved by the RL agent. This is likely due to an issue encountered in imitation learning where the distribution of data used to train the agent differs from the distribution of data encountered while testing the agent, since the agent's actions directly affect the future states it encounters. This leads to compounding errors and hurts the performance of the imitation learning agent (Ross & Bagnell, 2010). These low scores are also consistent with Kanervisto et al. (2020), which found that behavioral cloning from human gameplay was unable to match human-level gameplay scores for the Atari games considered in that paper.

### 4.4.3 Scalability of the Tensor Train

Differences in the speedup between the limited data regime and moderate data regime are due primarily to the number of training epochs used in training the AE/VAE - we use 50 training trajectories in the limited data regime and 5 in the moderate data regime. The reason for this difference is that we found that more training epochs were needed in the limited data regime to achieve similar accuracies via the AE/VAE.

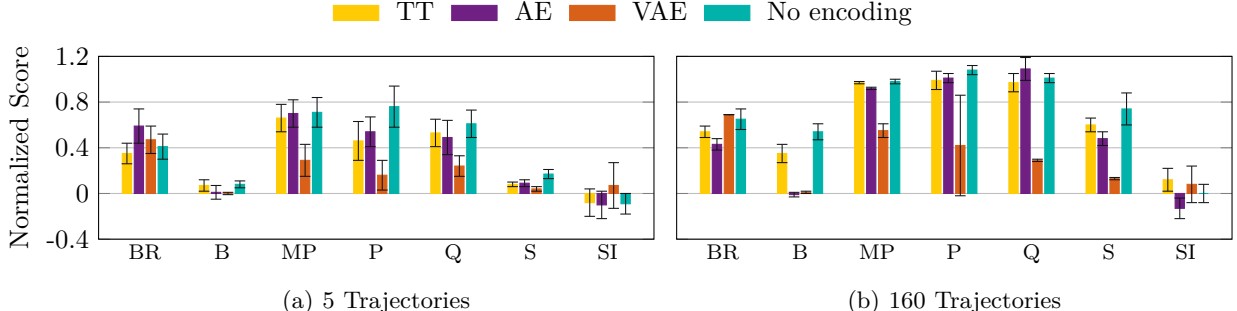

(a) 5 Trajectories

(b) 160 Trajectories

Figure 5: *Normalized gameplay scores achieved by each of the behavioral cloning models, trained using either encodings from the TT, AE, VAE, or the original RGB images. A score of 0 corresponds to performance of an agent taking random actions and a score of 1 corresponds to performance of the RL agent. Training a predictor using TT encodings yields similar gameplay scores as training a predictor using AE or VAE encodings. Error bounds are calculated as two times the standard error of gameplay scores calculated across ten (in the 5 trajectory case) or three (in the 160 trajectory case) experiments.*

Table 4: *Reconstruction quality of the TT, AE, and VAE in the moderate data regime. The reported statistics are the MSE (using 255 as maximum pixel value) and PSNR of the reconstructions versus the original frames. For each game, the bolded values are the smallest MSE and highest PSNR.*

|  | MSE | | | PSNR | | |
|---|---|---|---|---|---|---|
|  | TT | AE | VAE | TT | AE | VAE |
| BeamRider (BR) | **0.4** | 18.0 | 26.0 | **58.1** | 37.6 | 36.4 |
| Breakout (B) | **1.2** | 7.8 | 19.8 | **51.0** | 40.1 | 36.8 |
| MsPacman (MP) | **1.7** | 16.2 | 75.7 | **48.1** | 37.6 | 30.2 |
| Pong (P) | 1.8 | **0.0** | 6.9 | 47.0 | **71.4** | 40.7 |
| Qbert (Q) | **0.8** | 12.1 | 36.4 | **52.7** | 39.0 | 33.8 |
| Seaquest (S) | **1.0** | 8.4 | 52.9 | **49.5** | 39.9 | 31.4 |
| SpaceInvaders (SI) | **0.4** | 18.5 | 80.2 | **56.1** | 36.5 | 30.1 |

To make the speedups of the TT more comparable across different amounts of training data, we run a small study to examine the effect of increasing the number of gameplay trajectories used to train the behavioral cloning framework on both training time and prediction accuracy. The AE and VAE are trained using five epochs, matching the training setup used in the moderate data regime. We ran this for the game of Seaquest, given the relatively low prediction accuracies we achieved compared to some of the other Atari games. As expected, Figure 6 shows that the accuracy increases as more training data is used, suggesting that better prediction accuracies can be achieved with more data.

We note that TT's speedup over the AE / VAE and no encoding remains similar as more data is used, as long as the number of training epochs is fixed. These results suggest that the timing advantage may remain for the TT even when more data is available for behavioral cloning.

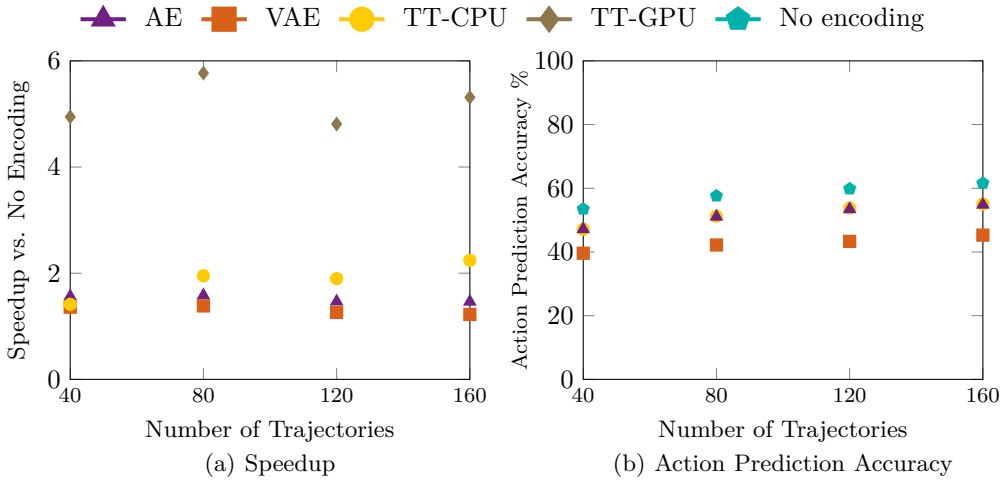

(a) Speedup  (b) Action Prediction Accuracy

Figure 6: *Results of the experiments investigating the effect of number of trajectories used to train the behavioral cloning framework on computational speedup and action prediction accuracy using different encoding methods for the game of Seaquest. As the number of training trajectories increases, action prediction accuracy also increases. Computational benefits of* `TT-ICE`* *over the AE, VAE, or no encoding remain similar across different number of trajectories. Action prediction accuracies are only presented for TT-CPU since TT-CPU and TT-GPU result in the same latent space.*

### 4.4.4  Prediction Accuracy Versus Training Time

A significant difference between training the AE / VAE versus the TT is the number of passes through each data point. Since the amount of training data is limited, multiple passes through the data are used for the AE/VAE to increase the number of update steps. On the other hand, the TT is trained using an algorithm that requires only a single pass through the data. As a result, the AE/VAE process more data points during training than the TT, albeit with repeating data.

To address this difference, we run experiments where each algorithm is trained with one epoch and as such each algorithm processes the same number of data points. The resulting accuracies and timings are plotted in Figure 7. As can be seen, TT-GPU tends to achieve higher accuracies and lower end-to-end training time than the AE/VAE.

### 4.4.5  Encoding Times and Inference Speed

During inference, new frames are passed through the TT, AE, or VAE, before the resulting latent vector is passed into a predictor. Because the architecture of the predictor is held fixed, differences in inference speed will stem from differences in the time it takes to encode frames using the various methods. These encoding times are compared in Table 5.

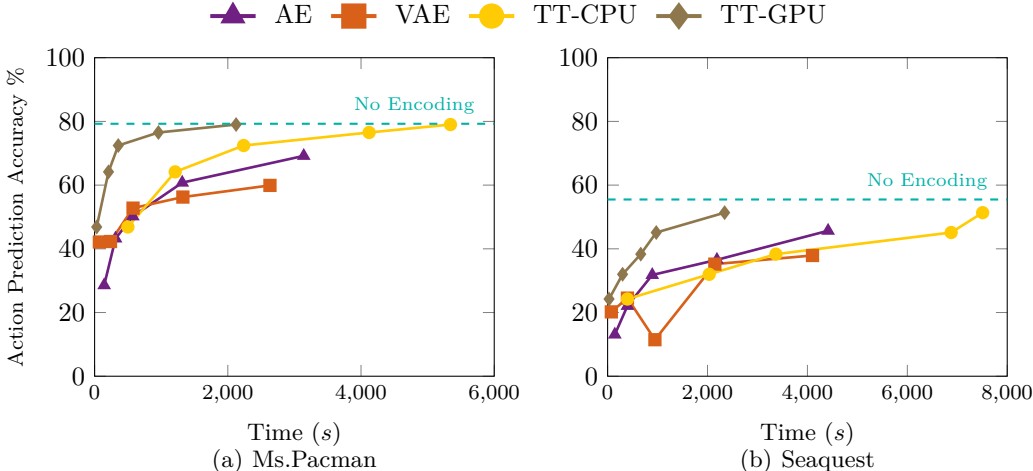

Figure 7: *Comparison of behavioral cloning accuracy versus end-to-end training time for the AE, VAE, and TT, when keeping the number of data points seen by each algorithm the same by using only a single pass through the data. Points in the upper left indicate lower training time and higher accuracy. Note that TT-GPU tends to perform better in terms of lower training time and higher accuracy than the AE and VAE.*

In the 5 trajectory case, the encoding time for TT-GPU is between 0.4× and 2.8× that of the AE and VAE. In the 160 trajectory case, the encoding time for TT-GPU is between 1.5× and 8.8× that of the AE and VAE. The current Python implementation of `TT-ICE`* is not optimized for an inference task like the one we have in this work. As shown in Algorithm 1, encoding new data using the TT-encoder is a task that is sequential in the dimensions. More efficient parallel approaches may be envisioned that can further accelerate the encoding task. There are multiple works in the literature on scaling such operations (Daas et al., 2022; Solomonik et al., 2014), and perhaps these can be considered in the future. However, even with this inefficiency, our results indicate faster training.

To determine why encoding is slower for the TT, we can investigate both flop counts and actual compute time. Moreover, we can see how these vary with dataset size. Flop counts (e.g., the number of multiply-add operations) for the AE and VAE are calculated using the Ptflops package (Sovrasov, 2023). Flop counts for the TT are calculated by counting the number of flops per matrix multiplication in Algorithm 1 (lines 9 and 13).

In Table 6, the ratio in encoding times and flop counts between the TT and AE/VAE are shown. Although the flop counts in the TT are generally higher, this does not lead to a proportional increase in encoding time (e.g., TT-GPU uses 5.8× as many flops as the AE in the 5 trajectory case for BeamRider, but the encoding time for TT-GPU is 2.0× as long as the encoding time for the AE). This is partly due to the number of sequential calculations needed for each encoding method; in our experiments, the AE/VAE requires thirteen sequential layers (intermediate layers, including batch norm layers and activation functions), while TT projection requires only five tensor contractions (the number of tensor cores minus one).

We also explore the scalability of the inference speed of the TT with the amount of data. During training, the tensor ranks $r$ of the TT increase as needed to represent new data. The number of flop counts for encoding using the TT grows quadratically with $r$. As such, the scalability of the inference speed of the TT depends on the ability of the TT to learn a generalizable latent structure that can accurately represent new data, as this will lead to fewer and smaller increases in $r$ as new data is introduced.

It is difficult to determine this scalability *a-priori*, but we empirically investigate the growth in the tensor ranks and flop counts in Figure 8. As the amount of data increases, the maximum rank of the TT grows, but the growth rate tends to level off. The slow growth of the TT ranks suggest that for these Atari games, the TT is able to learn a generalizable latent structure and scales well with additional data.

Table 5: *Median time to encode the training and validation data in seconds using the various encodings. In the 5 trajectory case, the encoding time for TT-GPU is between 0.4× and 2.8× that of the AE and VAE. In the 160 trajectory case, the encoding time for TT-GPU is between 1.5× and 8.8× that of the AE and VAE.*

|  | 5 Trajectories | | | | 160 Trajectories | | | |
|---|---|---|---|---|---|---|---|---|
|  | AE | VAE | TT-CPU | TT-GPU | AE | VAE | TT-CPU | TT-GPU |
| BeamRider | 3.6 | 3.6 | 160.9 | 7.2 | 156.9 | 139.9 | 1906.5 | 673.5 |
| Breakout | 2.2 | 0.9 | 29.7 | 0.9 | 27.9 | 26.1 | 165.1 | 54.8 |
| MsPacman | 2.9 | 2.9 | 111.8 | 7.9 | 89.4 | 90.9 | 1153.2 | 484.4 |
| Pong | 12.8 | 12.4 | 519.2 | 25.4 | 433.4 | 448.1 | 1994.3 | 680.5 |
| Qbert | 2.0 | 2.0 | 64.1 | 2.9 | 60.3 | 60.6 | 388.5 | 123.6 |
| Seaquest | 3.2 | 3.1 | 116.3 | 7.9 | 128.3 | 132.9 | 1714.4 | 619.4 |
| SpaceInvaders | 2.1 | 1.9 | 73.8 | 3.1 | 71.7 | 72.3 | 1387.0 | 632.6 |

Table 6: *Ratio of encoding times and flop counts between the TT and AE/VAE. Although the flop counts are much higher for the TT than the AE/VAE, this does not lead to a proportionate increase in encoding time.*

|  |  | 5 Trajectories | | 160 Trajectories | |
|---|---|---|---|---|---|
|  |  | TT-GPU / AE | TT-GPU / VAE | TT-GPU / AE | TT-GPU / VAE |
|  | BeamRider | 2.0 | 2.0 | 4.3 | 4.8 |
|  | Breakout | 0.4 | 1.0 | 2.0 | 2.1 |
| Encoding | MsPacman | 2.8 | 2.8 | 5.4 | 5.3 |
| Time | Pong | 2.0 | 2.0 | 1.6 | 1.5 |
| Ratio | Qbert | 1.5 | 1.5 | 2.0 | 2.0 |
|  | Seaquest | 2.5 | 2.5 | 4.8 | 4.7 |
|  | SpaceInvaders | 1.4 | 1.6 | 8.8 | 8.7 |
|  | BeamRider | 5.8 | 4.8 | 12.8 | 7.8 |
|  | Breakout | 0.8 | 0.8 | 3.9 | 3.0 |
| Flop | MsPacman | 7.6 | 6.5 | 12.7 | 8.0 |
| Count | Pong | 1.9 | 1.7 | 2.3 | 2.0 |
| Ratio | Qbert | 5.3 | 4.9 | 7.5 | 6.2 |
|  | Seaquest | 4.8 | 3.9 | 10.4 | 6.3 |
|  | SpaceInvaders | 5.0 | 4.5 | 16.3 | 9.7 |

An area of future research will be to improve the encoding time of the TT. However, we reinforce that though the encoding time of the TT is longer than that of the AE and VAE, the overall time for training the TT on GPU is still much lower than that of the AE and VAE since training the encoding is the most time-intensive task.

### 4.4.6 Additional Results

Appendix B.1 provides a more detailed investigation into the timing differences between the methods. The speedups achieved by the TT compared to the AE and VAE are split into three components:

1. Time to train the encoder.

2. Time to encode training and validation data used to train the action predictor.

3. Time to train the action predictor.

To ensure that our results are consistent across different settings, we also examine how changing the latent space size (Appendix B.2) affects the training time and action prediction accuracy of the AE, VAE, and TT. The results are consistent: we find that as the latent space size and number of training trajectories are changed, TT achieves similar prediction accuracies as the AE and VAE but provides a speedup in training time.

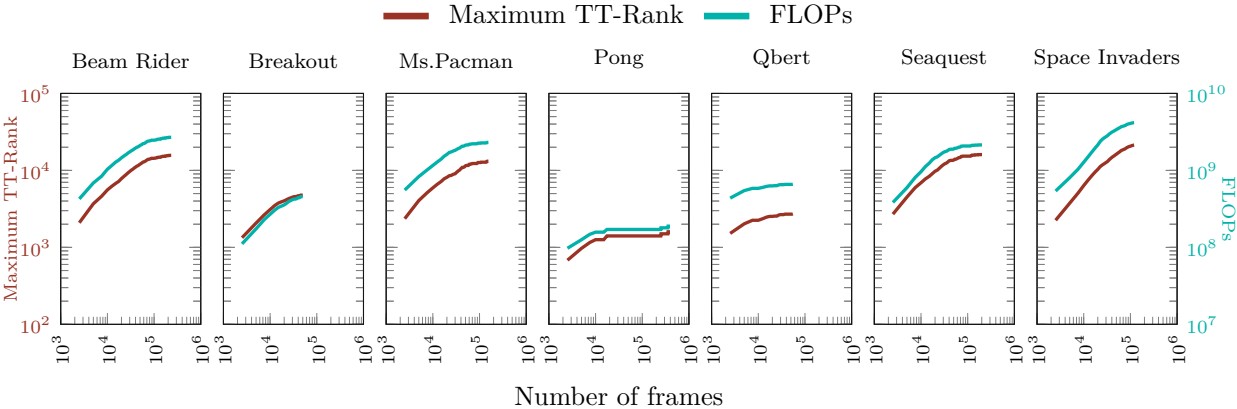

Figure 8: *Maximum tensor rank (red) and flop count (teal) for the TT during training. As training progresses, the maximum tensor rank and the flop count increase, but this growth rate decreases. The slow growth in the TT ranks suggests that the `TT-ICE*` algorithm is able to learn a generalizable latent structure; as the amount of data seen by the `TT-ICE*` algorithm increases, smaller updates to the TT ranks are needed to represent new data. In cases like this where the tensor ranks of the TT grow slowly, the encoding time for the TT will scale well since the flop count grows quadratically with the TT ranks $r$.*

In Appendix B.3, we provide the gameplay scores achieved by the RL agent used to generate the demonstration data we used for behavioral cloning. The RL agent achieves gameplay scores that are higher than a randomly acting agent and achieves human level performance in two of the seven games.

## 5    Conclusion

In this work, we have explored using tensor networks as a method for learning a low-dimensional representation of image data for behavioral cloning. In particular, we compare the computational speed of using TT against AEs and VAEs for extracting this low dimensional representation, finding a $2.6\times$ to $9.6\times$ speedup compared to AEs or VAEs. Furthermore, the TT-encoding can be efficiently trained on CPU as well, which leads to comparable or better training times than the AE and VAE trained on GPU ($0.9\times$ to $5.4\times$ faster). This speedup comes at no cost in terms of behavioral cloning accuracy or the gameplay scores achieved by the behavioral cloned agents. As such, the TT-based approach enables learning while having limited access to GPUs, as it allows for the expensive dimensionality reduction step to be performed on CPUs at a comparable or faster time than using an AE or VAE on GPUs. Given the large computational requirements of learning directly from video data, future work will aim to extend this methodology to tackling larger game environments for which the large input space is a computational bottleneck.

Although this paper explores the use of tensor networks for extracting a latent space from video data for behavioral cloning, the approach outlined in this paper may be broadly applicable. Tensor networks have been shown to successfully compress data in a wide variety of settings, ranging from EEG signals (Zhang et al., 2017), video data (Zhao et al., 2020; Fonał & Zdunek, 2019), images (Fonał & Zdunek, 2019), and audio (Fonał & Zdunek, 2019). Tensor networks can be generally applied to tensorizable features, as many data formats are naturally tensors (e.g., videos are 4-way tensors). One and two dimensional data can also be reshaped into tensors then compressed via TT. We also note the possibility of using TT to compress tensorizable features and using these compressions along with non-compressed features as input into a learning algorithm.

Abstractly, the benefits of this approach are to problems that have separable structure, where certain features of the problem interact in limited ways with others. Different tensor network structures define such interactions differently, based on network topology. It is difficult to *a-priori* say if a particular application exhibits this structure, and so the above literature and the current work continue to provide empirical evidence that

this structure exists in quite general problems. A future area of research would be further exploration of data and tasks for which the TT format works well as a method of feature extraction.

Another interesting research direction would be to explore if a TT-based encoder can be trained using frames from multiple games. If frames from different video games do share some common structure, it is possible that the rank of the TT would grow sublinearly with the number of games included, leading to a more storage-efficient alternative. It would also be interesting to explore if an encoder trained in this fashion would be able to generalize to similar but out-of-sample games.

### Acknowledgments

We acknowledge support from the Automotive Research Center (ARC) in accordance with Cooperative Agreement W56HZV-19-2-0001 with U.S. Army DEVCOM Ground Vehicle Systems Center. This research was supported in part through computational resources and services provided by Advanced Research Computing at the University of Michigan, Ann Arbor.

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

## Appendix

## A    Additional Details on Training

Architectural details for the encoding part of our AEs and VAEs are laid out in Table 7. The decoding portion of these networks mirrors the encoding portion, with convolutional layers replaced by transposed convolutional layers.

Table 7: *Architecture for the encoding part of AEs and VAEs. After each intermediate layer we also apply batch norm followed by LeakyReLU - these are excluded from the table for simplicity. The asymmetric padding in the first layer is to account for the shape of the input.*

| Layer # | Layer Type | Inp. Ch. | Out. Ch. | Kernel Size | Stride | Padding |
|---|---|---|---|---|---|---|
| 1 | Conv | 3 | 32 | 3 | 2 | (4, 1) |
| 2 | Conv | 32 | 64 | 3 | 2 | (1, 1) |
| 3 | Conv | 64 | 64 | 3 | 2 | (1, 1) |
| 4 | Conv | 64 | 16 | 1 | 1 | (0, 0) |
| 5 | Flatten | - | - | - | - | - |
| 6 | Linear | - | - | - | - | - |

The AEs and VAEs are hyperparameter tuned individually for each game over initial learning rates of $10^{-2}, 10^{-3}$, and $10^{-4}$ and L2 weight decay penalties of $1, 10^{-2}$, and $10^{-4}$, after which the best model is selected based on validation error and used for encoding. The networks are trained using the Adam-W optimizer (Loshchilov & Hutter, 2019) with cosine learning rate decay without restarts (Loshchilov & Hutter, 2017) for fifty epochs in the low data case and five epochs in the moderate data case. The AE is trained to minimize reconstruction error measured via mean squared error while the VAE is trained using the standard ELBO loss (Kingma & Welling, 2014). An additional 1 or 20 trajectories (in the limited and moderate data regimes, respectively) are used as a validation set in order to hyperparameter tune the networks. Note that the TT does not require this step of validation and does not use this additional data.

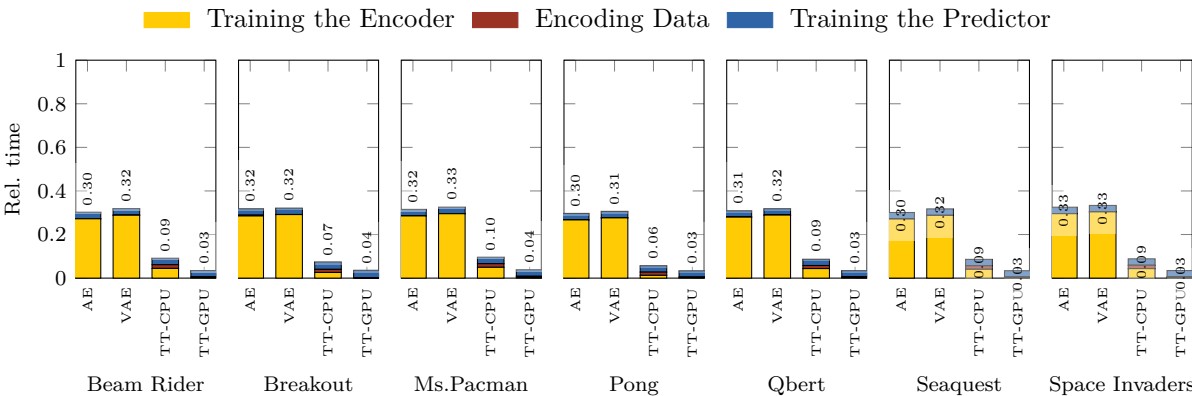

Figure 9: *Time for end-to-end training procedure in the limited data regime, averaged across all experiments, as a fraction of the time required to train the full-frame predictor. Extracting a latent space before training a predictor reduces end-to-end training time in all cases, but using the TT-GPU leads to a 26.1× to 29.7× reduction in training time. Compared to the AE / VAE, using the TT-GPU leads to a 8.2× to 9.6× speedup.*

## B  Additional Results

### B.1  Breakdown of Speedups

In Figures 9 and 10, we further break down the timings for training the AE, VAE, and TT-based predictors as follows:

- ***Training the encoder:*** For the AE and VAE, this time is the total time required to perform a hyperparameter optimization by training nine AEs / VAEs with different network hyperparameters and selecting one based on the validation loss measured on a separate validation set. Recall that the TT encoder does not require such a procedure. For the TT, this is the time required to perform the TT decomposition (using `TT-ICE`*) to extract the latent space representation of the input data.

- ***Encoding data:*** Time to pass all the training and validation data through the AE, VAE, and TT in order to extract latent space representations.

- ***Training the action predictor:*** Time to train the action predictor on the encodings extracted in the previous step. The architecture of the predictor is same across all three methods. For each model and for each game, nine predictors are trained using different initial learning rates ($10^{-2}$, $10^{-3}$, and $10^{-4}$) and L2 regularization penalties ($1$, $10^{-2}$, and $10^{-4}$). The predictor yielding the lowest cross-entropy action prediction loss measured on another separate validation set is selected.

Note that the most time-intensive task is training the encoding. As such, reducing the time it takes to train the encoding by using a TT instead of an AE or VAE leads to a significant reduction in overall training time.

### B.2  Effect of Changing Epsilon

We conduct experiments testing the effect of changing the size of the latent space for the TT, AE, and VAE. In our experiments, the latent size of the AE and VAE is set up to match the latent size of the TT. This experiment is conducted using 160 trajectories as training data, with five training epochs for the AE and VAE. In Table 8, the latent size as epsilon is changed is displayed.

In Figure 11, the effect of changing epsilon / the latent size on prediction accuracy is displayed. The general pattern appears to be that decreasing epsilon / increasing the latent size leads to a higher prediction accuracy, as expected. However, for some games, increasing the latent space size past a certain point leads to small or negligible improvements in prediction accuracy. The effect of changing the latent space size on timings

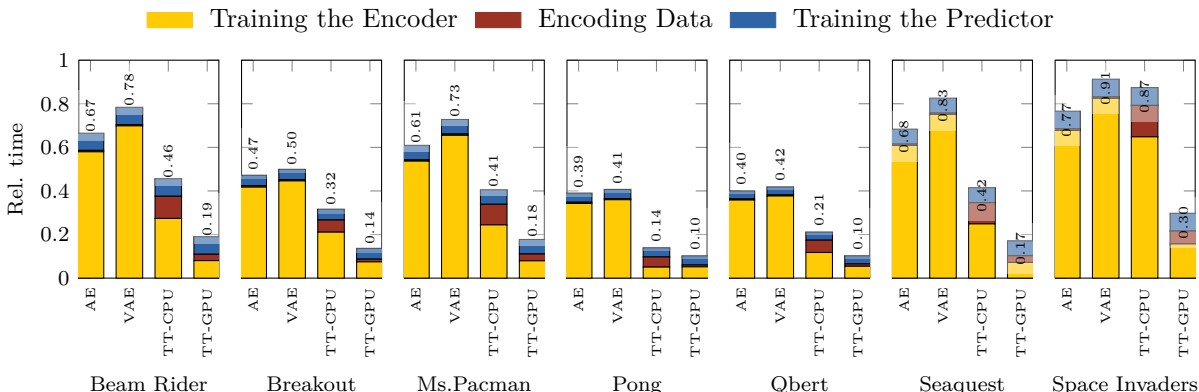

Figure 10: *Time for the end-to-end training procedure in the moderate data regime, averaged across all experiments, as a fraction of the time it takes to train a predictor using the full RGB frames. Extracting a latent space reduces end-to-end training time in all cases, but using TT-GPU leads to a 3.4× to 9.8× reduction in training time compared to no encoding and a 2.6× to 4.8× speedup relative to the AE / VAE.*

Table 8: *Latent size as epsilon for the TT is changed. The reported numbers are an average across three seeds (which use different training data).*

| Epsilon | 0.01 | 0.02 | 0.05 | 0.10 | 0.15 |
|---|---|---|---|---|---|
| BeamRider | 14645 | 9867 | 3324 | 297 | 72 |
| Breakout | 3561 | 957 | 50 | 18 | 3 |
| MsPacman | 11962 | 6421 | 867 | 15 | 2 |
| Pong | 1287 | 138 | 4 | 1 | 1 |
| Qbert | 2267 | 1560 | 308 | 43 | 10 |
| Seaquest | 15163 | 8170 | 1338 | 37 | 1 |
| SpaceInvaders | 17191 | 13795 | 8035 | 3994 | 2097 |

is shown in Figure 12. Using a larger latent space size tends to increase the end-to-end training time of the models. These results are consistent with our findings in the main text - at the different epsilon levels tested, the TT-based predictor yielded similar prediction accuracies as the AE and VAE, while being faster to train.

## B.3 Gameplay Performance of RL Agent

This section provides some statistics on the RL agent used to generate the demonstration data.

Table 9 provides the gameplay scores achieved by the RL agent compared to an agent that acts randomly by uniformly selecting amongst available actions and a human player. The human player scores are taken from Mnih et al. (2015). The RL agent achieves higher scores than an agent acting randomly and achieves human-level performance in 2 of the 7 games. Table 10 provides the gameplay lengths (number of steps between the beginning of a game to the end) achieved by the RL agent versus an agent that acts randomly. Note that gameplay lengths were not available from Mnih et al. (2015). This table indicates that the RL agent has learned behavior that differs from a randomly acting agent.

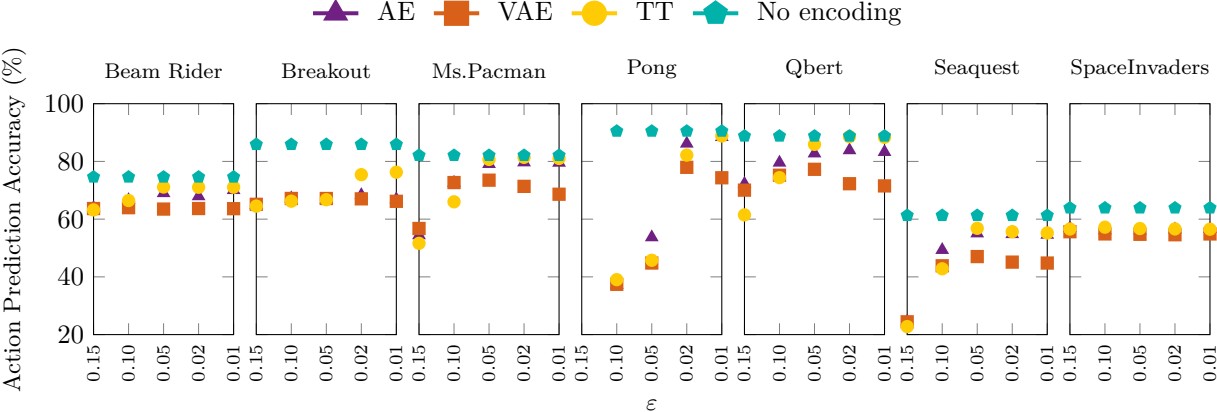

Figure 11: *Action prediction accuracy of behavioral cloning frameworks using different encoding methods at varying ε levels. As the latent space size is increased for the encoding-based predictors, the action prediction accuracy of the predictors tends to increase. The speedup offered by TT does not cause any performance deterioration. Furthermore, the nonlinearity offered by AE/VAE does not result in an improved latent space representation that achieves higher action prediction accuracy with a smaller latent space. Since Pong with ε = 0.15 and ε = 0.10 returns the same latent size, the accuracy for ε = 0.15 is omitted. Please refer to Table 8 for latent space sizes corresponding to each ε level for each game.*

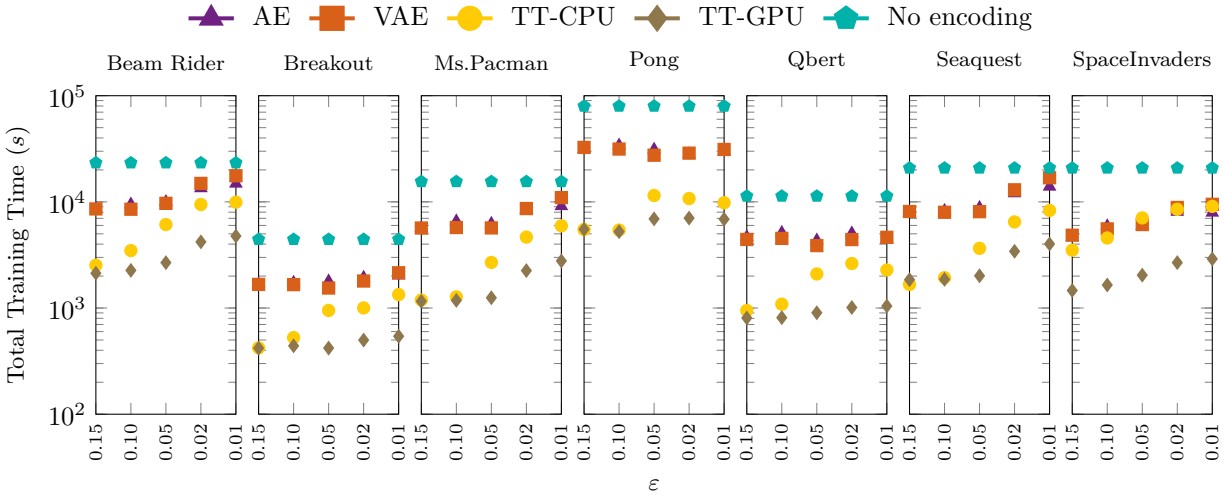

Figure 12: *Total time elapsed to train behavioral cloning frameworks using different encoding methods at varying ε levels. As the latent space size increases, the total training time for the various encoding-based predictors increases. Across all games and ε levels, TT-GPU encoding offers the fastest end-to-end training time. Please refer to Table 8 for latent space sizes corresponding to each ε level for each different game.*

Table 9: Comparison of the gameplay scores achieved by the RL agent compared to an agent that acts randomly and a human player. The human gameplay scores are taken from Mnih et al. (2015).

|  | Random | RL | Human (Mnih et al., 2015) |
|---|---|---|---|
| BeamRider | 366.1 | 1012.3 | 5775.0 |
| Breakout | 1.4 | 64.0 | 31.8 |
| MsPacman | 244.8 | 1883.4 | 15693.0 |
| Pong | -20.3 | 21.0 | 9.3 |
| Qbert | 167.5 | 5181.0 | 13455.0 |
| Seaquest | 67.2 | 1962.0 | 20182.0 |

Table 10: Comparison of the gameplay lengths (number of time steps from the beginning of the game to the end) achieved by the RL agent compared to an agent that moves randomly. Note that gameplay lengths for a human player were not available from Mnih et al. (2015).

|  | Random | RL |
|---|---|---|
| BeamRider | 1289.5 | 1714.3 |
| Breakout | 153.2 | 982.1 |
| MsPacman | 489.7 | 1025.0 |
| Pong | 926.8 | 1570.7 |
| Qbert | 308.8 | 1390.0 |
| Seaquest | 492.2 | 2414.9 |
| SpaceInvaders | 488.7 | 660.5 |

