# OpenReview forum: "Low-Rank Tensor-Network Encodings for Video-to-Action Behavioral Cloning"
_TMLR — Accepted by TMLR_

### Review · Reviewer_S6tk · 2024-01-20

**Summary Of Contributions:**

In this work, the authors leverage a tensor-train network as the feature encoder dealing with raw video data in the context of video-to-action behavior cloning. The use of TTN for feature extraction is interesting in the new applied scenarios, and the authors' experiments could verify the effectiveness of TTN.

**Audience:**

Yes

**Broader Impact Concerns:**

There is no ethical issue in this work.

**Claims And Evidence:**

Yes

**Requested Changes:**

(1) To improve the quality of this paper, more theoretical explanations are expected for something uncommon using TTN for the new application. It is unclear why the TTN can be effectively applied in video-to-action behavioral cloning.

(2) More low-rank tensor methods are expected to be attempted.

(3) More discussions on the efficiency of low-rank TTN on GPUs should be expected.

**Strengths And Weaknesses:**

Strengths:
1. The use of TTN as an encoder for low-dimensional feature extraction is interesting in the scenario of image data for behavior cloning.

2. The TTN exhibits some performance gain in the experiments compared with AEs and VAEs.

3. The discussion of the advantages of using TTN on GPUs is an important new contribution.

Weaknesses:
1. The technical novelty of this work is limited. The TTN model was proposed by Oseledets in 2011, and it has already been applied to speech, image, and signal processing. In particular, Yang et al., (2017) put forth using the TTN for video data processing, and Qi et al., (2023) even used low-rank Tensor-Train Network for speech processing.

[1] I. V. Oseledets, “Tensor-train decomposition,” SIAM J. Sci. Comput., vol. 33, no. 5, pp. 2295–2317, 2011.

[2] Y. Yang, D. Krompass, and V. Tresp, “Tensor-train recurrent neural networks for video classification,” in Proc. Int. Conf. Mach. Learn., 2017, pp. 3891–3900.

[3] Qi, J., Yang, C.H.H., Chen, P.Y. and Tejedor, Exploiting Low-Rank Tensor-Train Deep Neural Networks Based on Riemannian Gradient Descent With Illustrations of Speech Processing. IEEE/ACM Transactions on Audio, Speech, and Language Processing, 31, pp.633-642, 2023.

2. The authors do not show theoretical support to explain the low-rank TTN for the successful application in this work. The authors can refer to the recent theoretical work of Qi et al. on TTN as an encoder exhibiting low-dimensional representations of input features, where many theoretical bounds are provided in the contexts of end-to-end TTN and pre-trained TTN models.

[4] Qi, J., Yang, C.H.H., Chen, P.Y. and Hsieh, M.H., Theoretical error performance analysis for variational quantum circuit based functional regression. npj Quantum Information, 9(1), p.4, 2023.

[5] Qi, J., Yang, C.H.H., Chen, P.Y. and Hsieh, M.H., Pre-training Tensor-Train Networks Facilitates Machine Learning with Variational Quantum Circuits. arXiv preprint arXiv:2306.03741, 2023.

3. The title is "low-rank tensor-network encoding," but the authors only discuss the TTN for feature extraction. If the authors use the tensor network, the authors need to compare other tensor decomposition methods like CP and Tucker decompositions, e.g., the CP decomposition for CNN below:

[6] Lebedev, V., Ganin, Y., Rakhuba, M., Oseledets, I. and Lempitsky, V., 2014. Speeding up convolutional neural networks using fine-tuned cp-decomposition. arXiv preprint arXiv:1412.6553.

4. The speed-up achieved by TTN on GPUs is not fully explained. Different setups of TT ranks can greatly affect the running speed but with a compromise of lowering the empirical performance. The compromise of the efficiency of TTN and its empirical performance should be further discussed in both theory and experiments.

---

> ### Author Response · Authors · 2024-02-13
>
> Thank you for your comments. In the below, we address your concerns laid out in “Weaknesses”.
> Please let us know if you have further questions.
>
> We have also made edits to the paper, which are described in the responses. Note that we
> used blue for edits in response to S6tk, red for edits in response to k7Jk, and magenta for edits in
> response to wupi.
>
> >The technical novelty of this work is limited. The TTN model was proposed by Oseledets in
> 2011, and it has already been applied to speech, image, and signal processing. In particular,
> Yang et al., (2017) put forth using the TTN for video data processing, and Qi et al., (2023)
> even used low-rank Tensor-Train Network for speech processing.
> – I. V. Oseledets, “Tensor-train decomposition,” SIAM J. Sci. Comput., vol. 33, no. 5,
> pp. 2295–2317, 2011.
> – Y. Yang, D. Krompass, and V. Tresp, “Tensor-train recurrent neural networks for video
> classification,” in Proc. Int. Conf. Mach. Learn., 2017, pp. 3891–3900.
> – Qi, J., Yang, C.H.H., Chen, P.Y. and Tejedor, Exploiting Low-Rank Tensor-Train Deep
> Neural Networks Based on Riemannian Gradient Descent With Illustrations of Speech
> Processing. IEEE/ACM Transactions on Audio, Speech, and Language Processing, 31,
> pp.633-642, 2023.
>
> We certainly agree that we did not invent the tensor-train ansatz. It has been around for a
> long time, even prior to 2011 by the name of Matrix Product States. The novelty of this work
> does not center on the proposal of a new tensor network ansatz. Instead, it provides empirical
> support for the idea that the complexity of representation provided by tensor networks may
> be sufficient for tasks currently dominated by deep neural network approaches. This support
> is newly made possible by incremental algorithms (such as TT-ICE [1]), which provide the
> basis for memory efficient training of these models (in contrast to TT-SVD of Oseledets
> 2011, or optimization-based incremental approaches). These new training methods do not
> require optimization, and they provide guarantees on reconstruction error. We contrast this
> contribution with that of Yang et. al. and Qi et. al. mentioned above. Those approaches
> embed tensor networks within neural networks to accelerate or compress the neural network
> approach. As such, they don’t mitigate the issues surrounding the complexity of optimization
> approaches. We show that a global tensor network is sufficient. A discussion is added to
> Section 2.
>
> > The authors do not show theoretical support to explain the low-rank TTN for the successful
> application in this work. The authors can refer to the recent theoretical work of Qi et al.
> on TTN as an encoder exhibiting low-dimensional representations of input features, where
> many theoretical bounds are provided in the contexts of end-to-end TTN and pre-trained TTN
> models.
> – Qi, J., Yang, C.H.H., Chen, P.Y. and Hsieh, M.H., Theoretical error performance anal-
> ysis for variational quantum circuit based functional regression. npj Quantum Informa-
> tion, 9(1), p.4, 2023.
> – Qi, J., Yang, C.H.H., Chen, P.Y. and Hsieh, M.H., Pre-training Tensor-Train Net-
> works Facilitates Machine Learning with Variational Quantum Circuits. arXiv preprint
> arXiv:2306.03741, 2023.
>
> We are not precisely clear on the types of theoretical support that the reviewer is looking for.
> However, we mention that there are two types of theoretical guarantees that one may make:
> guaranteeing reconstruction accuracy for a specific set of data and guaranteeing representation
> quality for a certain class of problems. With respect to the former, we are indeed able to
> guarantee reconstruction up to a given accuracy (see Theorem 2 and Theorem 4 in [1]). We
> empirically demonstrate results for the Atari gamesets for various error tolerances throughout
> the paper. The latter sort of theoretical justification is more difficult to address. If a problem
> can be a-priori shown to require a particular TT rank, then we should be able to say something
> about how much data is required to ensure accurate representation for a problem. However,
> these gamesets are complex and we do not envision one can apriori determine a useful upper-
> bound on the rank in the general case. Nevertheless, in the asymptotic limit of the data, our
> methods will be able to represent any tensor because of the rank adaptation properties. A
> brief discussion is added to Section 3.1.3.
>
> (response continued)

---

> ### Author Response · Authors · 2024-02-13
>
> (response continued from above)
>
> >The title is ”low-rank tensor-network encoding,” but the authors only discuss the TTN for
> feature extraction. If the authors use the tensor network, the authors need to compare other
> tensor decomposition methods like CP and Tucker decompositions, e.g., the CP decomposition
> for CNN below:
> – Lebedev, V., Ganin, Y., Rakhuba, M., Oseledets, I. and Lempitsky, V., 2014. Speed-
> ing up convolutional neural networks using fine-tuned cp-decomposition. arXiv preprint
> arXiv:1412.6553.
>
> Indeed we only use the tensor network for the encoding portion of the architecture — the
> overall framework for separating encoding and prediction is well established. However, we
> respectfully disagree that the above comparisons are required. First, the referenced paper
> is another approach for embedding a method within a neural network, and, second, many
> related incremental compression methods are optimization based. With respect to elaboration
> of these points, please see our answer to the first question above. We reinforce that removing
> the need for optimization-based method is the key reason for the acceleration that we see.
> The combination of the TT-format and the TT-ICE compression approach offers an attractive
> tensor network approach as it does not rely on optimization based algorithms for training (like
> CP-format may) or does not grow exponential with the number of dimensions (like Tucker
> format does). In addition to those advantages, we find that the literature lacks efficient
> incremental algorithms that provide theoretical error guarantees for the entirety of the training
> data, like the TT-ICE approach.
>
> >The speed-up achieved by TTN on GPUs is not fully explained. Different setups of TT ranks
> can greatly affect the running speed but with a compromise of lowering the empirical perfor-
> mance. The compromise of the efficiency of TTN and its empirical performance should be
> further discussed in both theory and experiments.
>
> First we reinforce that the overall speedup is due to a fast incremental approach for compressing the data, that does not require optimization. We adapt this approach to the GPU
> in a trivial way. The TT-CPU python implementation is modified to perform the linear algebra operations on GPUs with CUDA capabilities by converting the NumPy functions to
> CuPy functions. No other algorithmic changes were made to the original TT-ICE algorithm
> presented in [1]. A clarification on this is added to the manuscript.
>
> With respect to the TT-rank setups, we reinforce that the TT-ranks are automatically determined by the TT-ICE algorithm depending on the relative error threshold prescribed (1% for the experiments in the manuscript). According to the error threshold, the TT-ICE algorithm computes the optimal approximation (in the sense of TT-ranks). As the TT-ranks
> are automatically determined by the TT-ICE algorithm, there exists a tradeoff between the
> prescribed error threshold ε and efficiency of the TT. This tradeoff is explored in Appendix
> A.4. In Table 7, we present the size of the latent space (the final TT rank) as ε is changed.
> Furthermore, in Figures 10 and 11, we compare the action prediction accuracy and training
> time as ε is changed. These figures illustrate the tradeoff that occurs - as ε is decreased,
> action prediction accuracy tends to increase for the TT but training time also increases.
>
> [1] Doruk Aksoy, David J Gorsich, Shravan Veerapaneni, and Alex A Gorodetsky. An incremental
> tensor train decomposition algorithm. arXiv preprint arXiv:2211.12487, 2022.

---

### Review · Reviewer_k7Jk · 2024-01-23

**Summary Of Contributions:**

This paper aims to improve the computational efficiency of behavioral cloning. For extracting low-dimensional latent representations, it proposes an efficient dimension reduction method that is based on tensor decomposition.
- The behavioral cloning task is instantiated as action prediction in gameplays.
- The model used for action prediction consists of 2 parts: an encoding algorithm, and an action predictor.
  - The action predictor is a standard MLP, which is kept the same for different encoders.
- The main contribution of the paper lies in the choice of encoder: The paper proposed to use a tensor decomposition method to obtain a low-dimensional latent. The method of choice is TT-ICE, which is an incremental version of TT-SVD. The proposed TT-based method is better than baseline methods using AE/VAE, in terms of both computation time and performance.

Here are some more detailed results:
- Training time:
    - when using limited data (i.e. 5 trajectories per game): the TT-based method is around 9 times faster than AE/VAE, and around 28 times faster than predicting directly on pixels.
    - when using moderate data (i.e. 160 trajectories per game): the TT-based method is around 3 times faster than AE/VAE.
    - for both data regime, when the TT-based method is ran on CPUs, it is still faster than the other methods trained on GPUs.
- Performance:
    - pSNR and MSE: TT-based method is significantly better.
    - action prediction accuracy: TT-based method is mostly better, except for the game S and SI under limited data.
    - normalized gameplay scores: TT-based method is comparable with AE; both are better than VAE.
- Encoding time (related to inference time): TT-based method is no better.

**Audience:**

Yes

**Broader Impact Concerns:**

There are no direct ethical concerns.

**Claims And Evidence:**

Yes

**Requested Changes:**

- Please consider including more discussions on training scalability, generalizability, inference cost, and other design choices, as mentioned in weakness.
- Please consider comparing with more baselines, such as contrastive learning or masked prediction.
- There's a minor typo: Sec 4.1: "Rather than use" -> "Rather than using"

**Strengths And Weaknesses:**

Strengths:
- This proposed method is both conceptually elegant and empirically effective.
- The experiments include ablations, such as the latent space size and the amount of training data.
- The paper is clearly written and provides implementation details.

Weaknesses: I'm mainly concerned about the following aspects:
- **Training scalability**: It is noted in the paper that the main advantage of the TT-based method is the saving in the training time. However, by comparing the limited data regime with the moderate data regime, we see that the amount of speedup decreases as more data is used during training. This raises a question: how well does the TT-based method work as the data size further scales up?
- **Memory use and generalizability**: The method learns game-specific representations, i.e. the TT-cores. This means that the number of parameters grows linearly in the number of games, and the TT-based encoder cannot generalize to unseen games.
- **Inference cost**: the TT-based method's encoding time is high. The paper leaves this as a future direction, but I'm concerned that this will be a main bottleneck in adapting the proposed method in more general settings.
- Related to the above two points, it might be better if the authors could add some discussions on the applicability of their method. For example, for what kind of data do we expect an advantage of the proposed method?
- Several design choices could be better explained:
    - Choices of encoder baselines? For example, contrastive methods and masked prediction have been shown to produce better representations than (V)AE on images.
    - Choices of tasks: why these choices of Atari games? Disclaimer that I'm not an expert in RL so I apologize if this is a silly question.
    - Choices of the data generation process: currently DQN is used; (how) would the results be different as the data sources change?

---

> ### Author Response · Authors · 2024-02-13
>
> Thank you for your comments. In the below we address your comments - please let us know if you
> have any followup questions.
>
> We have also made edits to the paper, which are described in the responses. Note that we
> used blue for edits in response to S6tk, red for edits in response to k7Jk, and magenta for edits in
> response to wupi.
>
> >Training scalability:It is noted in the paper that the main advantage of the TT-based method
> is the saving in the training time. However, by comparing the limited data regime with the
> moderate data regime, we see that the amount of speedup decreases as more data is used during
> training. This raises a question: how well does the TT-based method work as the data size
> further scales up?
>
> We have added results to further explore the relationship between the amount of data used by
> the AE, VAE, and TT and the resulting behavioral cloning accuracy. This new experiment,
> described in Section 4.6, shows that TT-GPU tends to achieve higher accuracies and lower
> end-to-end training times than the AE and VAE (see Figure 7). In our main experiments, the
> TT is trained using a single pass through the training data via the TT-ICE algorithm without
> the need for optimization. On the other hand, the AE and VAE are trained using several
> training epochs (50 in the limited data case and 5 in the moderate data case) to increase the
> number of parameter updates during optimization. This effectively trains the AE/VAE with
> more “views” of the data, albeit repeated data. The experiment in Figure 7 controls this by
> training the AE, VAE, and TT with a single pass through the data.
>
> We have also moved results out of the Appendix into the main text comparing the training
> scalability over different numbers of training trajectories (see Section 4.5 and Figure 6). In
> these experiments, we fix the number of training epochs for the AE / VAE at 5 and the
> TT at 1. We emphasize that the TT does not require an optimization procedure and that
> the TT-SVD / TT-ICE algorithms allow the TT to be trained in a single pass through the
> data. Figure 6 shows that when the number of training epochs is held fixed for the game of
> Seaquest, TT’s speedup over the AE / VAE and no encoding remains similar as more data is
> used.
>
> >Memory use and generalizability: The method learns game-specific representations, i.e.
> the TT-cores. This means that the number of parameters grows linearly in the number of
> games, and the TT-based encoder cannot generalize to unseen games.
>
> We interpret this comment as a suggestion that one may be able to compress all different
> types of games simultaneously to achieve a “foundation model” sort of encoder for games.
> Furthermore, the comment suggests that this would not be efficient because of a linear growth
> in the parameter size as a function of the number of games.
>
> Given this interpretation, we would like to clarify that the representation does not scale as a
> direct function of number of games or data points. It is not a non-parametric representation
> in that sense. Overall, it is not clear what type of scaling would occur (sub-linear, linear,
> polynomial, etc.) Generally, the number of parameters in the encoders scales as $O(dnr^2)$,
> where $d$ is size of the features, $n$ is the size of each feature, and $r$ is the rank. If a common
> set of features can be identified across all games such that $d$ is fixed, then the number of
> parameters will scale quadratically with rank. If the games are related in a way such that the
> rank is bounded, then there would be no linear scaling. However, if the games are related in
> a way that increases the rank – then the number of parameters can grow faster than linear.
>
> As future work, it would be interesting to investigate the growth of TT-ranks when the TT-
> encoder is trained using images from multiple games. It would also be interesting to explore
> how well such an encoder extrapolates to other Atari games outside of its training set. If
> the frames from different games do share a common structure, it is possible that the rank
> of the TT grows sublinearly with the number of games included, which would offer a more
> storage-efficient alternative. We added a paragraph to the conclusions to point towards those
> possible future directions.
>
> (continued)

---

> ### Author Response · Authors · 2024-02-13
>
> (response continued from above)
>
> > Inference cost: the TT-based method’s encoding time is high. The paper leaves this as a
> future direction, but I’m concerned that this will be a main bottleneck in adapting the proposed
> method in more general settings.
>
> The current python implementation of the TT-ICE is not optimized for an inference task like
> the one we have in this work. As it is show in Algorithm 1 from our manuscript, encoding
> new data using the TT-encoder is a task that is sequential in the dimensions. More efficient
> parallel approaches may be envisioned that can further accelerate the encoding task. There
> are multiple works in the literature on scaling such operations [1, 5], and perhaps these can
> be considered in the future. We did not include it here because, even with this inefficiency,
> the results indicate faster training.
>
> >Related to the above two points, it might be better if the authors could add some discussions
> on the applicability of their method. For example, for what kind of data do we expect an
> advantage of the proposed method?
>
> We believe that the approach described in this paper is widely applicable, as tensor networks
> have been shown to successfully compress data in a wide variety of settings, ranging from
> EEG signals [6], video data [7, 2], images [2], and audio [2]. Given this ability to compress
> data, it is likely that the extracted features can be used as input to a neural network (which
> [7] does, but not the other papers mentioned in the previous sentence). However, this is not
> something we have tested.
>
> Abstractly, the benefit of this approach are to problems that have separable structure, where
> certain features of the problem interact in limited ways with others. Different tensor network
> structures define such interactions differently, based on network topology. It is difficult to
> apriori say if a particular application exhibits this structure, and so the above literature and
> the current work continue to provide empirical evidence that this structure exists in quite
> general problems. We have added a discussion of this to the conclusions.
>
> > Choices of encoder baselines? For example, contrastive methods and masked prediction
> have been shown to produce better representations than (V)AE on images.
>
> In our experiments, we focused on convolutional AEs / VAEs since, like the tensor train, these
> methods aim to learn a latent space from which the original images can be reconstructed. In
> future work, we hope to conduct further analysis comparing the TT to a wider range of
> neural network encoding approaches, such as ones based on contrastive learning and masked
> prediction.
>
> We note that because contrastive learning and masked prediction are a means of training a
> neural network encoding using a different loss / auxiliary task, the timings of using contrastive
> learning / masked prediction are likely to be similar to that of the AE / VAE, assuming similar
> neural network architectures. As such, timing results are likely to be similar, but as you point
> out, it is possible that this could lead to better encodings – primarily because of an augmented
> objective function for learning. This is a good point and other objectives for low-rank encoding
> is something we plan to explore in the future.
>
> From a practical standpoint, we do note that the behavioral cloning accuracy gap between
> using AE/VAE/TT encodings and using full frames is small for most of the games we explored,
> so using other encoding methods would likely yield similar behavioral cloning accuracies.
>
> > Choices of tasks: why these choices of Atari games? Disclaimer that I’m not an expert in
> RL so I apologize if this is a silly question.
>
> We selected these Atari games because trained reinforcement learning agents were available
> via the RL Baselines Zoo package [4]. The package also provided tuned hyperparameters
> for further training these models. We have added a sentence in Section 4.1 to mention this.
> We note that this package also provides trained reinforcement learning agents for the game
> Enduro. We have tested Enduro, but due to computational constraints of training the AE /
> VAE, we do not have results for AE / VAE for this game. In the moderate date case, the AE
> / VAE took roughly 30 hours to train per seed compared to 4.9 hours for the TT on GPU.
> We added a footnote to Section 4.1 with the above information.

---

> ### Author Response · Authors · 2024-02-13
>
> > Choices of the data generation process: currently DQN is used; (how) would the results
> be different as the data sources change?
>
> In our experiments, we used DQN since this reinforcement learning algorithm has been widely
> studied and has been shown to perform well on Atari games [3]. Furthermore, tuned hyper-
> parameters for training these models were readily available via the RL Baselines Zoo package
> [4]. A sentence has been added to Section 4.1 to explain the choice of DQN. We believe that
> our results would be similar if data were generated using a different reinforcement learning
> algorithm, as our main findings deal with being able to mimic any agent rather than consider
> some sort of optimal performance.
>
> [1] Hussam Al Daas, Grey Ballard, and Peter Benner. Parallel algorithms for tensor train arith-
> metic. SIAM Journal on Scientific Computing, 44(1):C25–C53, 2022.
>
> [2] Krzysztof Fonal and Rafal Zdunek. Distributed and randomized tensor train decomposition for
> feature extraction. In 2019 International Joint Conference on Neural Networks (IJCNN), pages
> 1–8. IEEE, 2019.
>
> [3] Volodymyr Mnih, Koray Kavukcuoglu, David Silver, Andrei A. Rusu, Joel Veness, Marc G.
> Bellemare, Alex Graves, Martin Riedmiller, Andreas K. Fidjeland, Georg Ostrovski, Stig Pe-
> tersen, Charles Beattie, Amir Sadik, Ioannis Antonoglou, Helen King, Dharshan Kumaran, Daan
> Wierstra, Shane Legg, and Demis Hassabis. Human-level control through deep reinforcement
> learning. Nature, 518:529–533, 2015.
>
> [4] Antonin Raffin. Rl baselines zoo. https://github.com/araffin/rl-baselines-zoo, 2018.
>
> [5] Edgar Solomonik, Devin Matthews, Jeff R Hammond, John F Stanton, and James Demmel. A
> massively parallel tensor contraction framework for coupled-cluster computations. Journal of
> Parallel and Distributed Computing, 74(12):3176–3190, 2014.
>
> [6] Wenjie Zhang, Jiqing Han, and Shiwen Deng. Heart sound classification based on scaled spectrogram and tensor decomposition. Expert Systems with Applications, 84:220–231, 2017
>
> [7] Bin Zhao, Xuelong Li, and Xiaoqiang Lu. Tth-rnn: Tensor-train hierarchical recurrent neural
> network for video summarization. IEEE Transactions on Industrial Electronics, 68(4):3629–
> 3637, 2020.

---

### Review · Reviewer_wupi · 2024-01-30

**Summary Of Contributions:**

This paper explores the use of tensor-network representations (a multi-linear decomposition & approximation to the data) for behavior cloning, in particular as a replacement to AE or VAE representations. The scope and claims are mostly about speed gains from using tensor network representations. They demonstrate this empirically on a few ATARI games.

Overall the paper is very clear about what it intends to do, and what it claims is very sharply and tactically defined. In effect most of my questions are more about what to expect from TT formats and comparing to more recent deep learning methods. But given the paper really only claims computational gains and not much more, I feel like it passes the TMLR two questions test for me, and hence I lean towards acceptance. I feel like people in the TMLR community will benefit from learning about the TT format.

**Audience:**

Yes

**Broader Impact Concerns:**

No concerns

**Claims And Evidence:**

Yes

**Requested Changes:**

1. Move requested hyperparameters/choices of (1) earlier in the main text
2. Discuss low performance in Figure 4 and 5 to ensure they do not affect applicability of these results

**Strengths And Weaknesses:**

1. The paper is clearly written, presents the scope, work and all components well. It presents earlier work very clearly. It is only in subtle situations where I think the paper could use more details:
   1. How L is set for AE/VAE (only appears in 4.2)?
   2. What N is (I assume 2500 given 3.1)?
   3. How the action prediction accuracy is computed (I assume it is likelihood of RL agent under learned action logits)
   4. The choice of using a 6-way TT format, and how each individual dimension was selected aren’t clear.
2. The comparison to AE and VAE baselines was well executed and overall the emphasis being put on speed comparisons given similar accuracies was useful.
   1. One weakness is the particular choice of VAE selected. Nowadays, SOTA results on discrete domains like ATARI are usually leveraging discrete generative representations (either DAE, VQ-VAE, VQ-GAN, etc, see DreamerV3) or contrastive representations (usually VIT based, see MineClip).
   2. Would it have led to a situation where the learnt encoder would have a higher accuracy?
3. The results in Figure 4 and 5 appear quite low.
   1. Could you comment on the accuracies and if they arise from specifically hard trajectories to predict, or if they are just symptoms of under-fitting BC policies?
   2. Similarly, Figure 5 has several games at chance or worst, what would be causing this?
   3. It is also hard to tell what the actual ability level of the base RL agents are. How do the raw scores obtained on each game relate to SOTA agents on Atari? Are they human-level? Are the trajectories representing interesting and complex behaviors?
   4. Figure 4 and 5 both seem to show that using raw RGB (“no encoding” baseline) are the best options. This usually isn’t the case when there is some transfer gap, could you comment on that?
4. The setup assumes frame-level encoding, followed by an action predictor which just consumes 4 frames to predict the current action.
   1. Did you explore a larger number of frame stacking or other architectures for the action predictor?
   2. How would one integrate the two and leverage videos in the representation?
5. Most of my worries about the paper are really about the TT format and its implications. Given this isn’t as relevant to this particular paper given its claims, I am not expecting these to be addressed, but would be interested in hearing the authors’ thoughts on them:
   1. Isn’t a multilinear decomposition too “weak” to represent complex visual scenes in a way that supports generalization? I would have expected non-linearities to help, compared to “just” doing SVD?
   2. Does the choice of dimensions (i.e. the reshaping done on the 4-dims videos to get to the 6-dim matrix) impact the performance/accuracy of the TT approximation or what it focuses on?
   3. You mention that TT-SVD is limited to 30’000 frames. Does TT-ICE* have some kind of limitations (e.g. is the accuracy on earlier datapoints reduced as N increases?). Would it scale to O(1M) frames (imagine using the MineCLIP dataset)?
   4. How would you extend it to handle videos in a way that shares information across the time axis? As far I understand, G6 is independent across frames?

---

> ### Author Response · Authors · 2024-02-13
>
> Thank you for the comments. In the below, we address your comments. Please let us know if you
> have any followup questions.
>
> We have also made edits to the paper, which are described in the responses. Note that we
> used blue for edits in response to S6tk, red for edits in response to k7Jk, and magenta for edits in
> response to wupi.
>
> > The paper is clearly written, presents the scope, work and all components well. It presents
> earlier work very clearly. It is only in subtle situations where I think the paper could use more
> details:
> – How L is set for AE/VAE (only appears in 4.2)?
> – What N is (I assume 2500 given 3.1)?
> – How the action prediction accuracy is computed (I assume it is likelihood of RL agent
> under learned action logits)
> – The choice of using a 6-way TT format, and how each individual dimension was selected
> aren’t clear.
>
> We have edited the paper to clear up these points of confusion. Specific answers to the
> questions are also provided below:
>
> – For the AE/VAE, we first train the TT. The latent size used for the AE/VAE is then
> set to be equal to that of the TT, in order to provide a fair comparison. We have
> also experimented with changing the latent size of the AE/VAE/TT, with results of
> these experiments in Appendix A.4, where we explore the tradeoff between increasing
> the latent size of these encoders - a larger latent size tends to lead to better behavioral
> cloning accuracies, at the cost of a longer training time. We’ve added a clarification to
> Section 4.2.
>
> – $N$ is indeed 2500. This has been added to Sections 3.1.3 and 4.2.
>
> – The action prediction accuracy is computed by calculating the proportion of times the
> action with the highest output probability from the various behavioral cloned agents
> is equal to the action with the highest output probability from the RL agent. We’ve
> clarified this in the main text in Section 4.3.
>
> – Reshaping the image files from $210 \times 160 \times 3$ to 6 dimensional tensors with dimensions
> $O(10)$ result in a more controlled growth in the intermediate TT-ranks. There’s a tradeoff
> between the number of dimensions and computational speed. Increasing the number of
> dimensions decreases the size of individual dimensions and therefore limits the growth
> of TT-ranks; however, as we need to perform SVDs sequentially for each TT-core during
> TT-ICE*, this increases the computation and inference time. We added a brief discussion
> to Section 3.1.3 on this.
>
> > The comparison to AE and VAE baselines was well executed and overall the emphasis being
> put on speed comparisons given similar accuracies was useful.
> – One weakness is the particular choice of VAE selected. Nowadays, SOTA results on
> discrete domains like ATARI are usually leveraging discrete generative representations
> (either DAE, VQ-VAE, VQ-GAN, etc, see DreamerV3) or contrastive representations
> (usually VIT based, see MineClip).
> – Would it have led to a situation where the learnt encoder would have a higher accuracy?
>
> Thank you for pointing us towards more SOTA results. These comparisons would be inter-
> esting and we hope to address this in future work.
>
> We note that because contrastive learning trains the neural network using a different auxilliary
> task, the timings of using contrastive learning are likely to be similar to that of the AE /
> VAE, assuming similar neural network architectures. Similarly, using discrete generative
> representations will also probably lead to similar timings.
>
> As such, the timing results are likely to be similar, but as you point out, it is possible that
> this could lead to better encodings and higher behavioral cloning accuracy. This is a good
> point and comparisons against SOTA approaches is something we would like to explore in
> the future. From a practical standpoint, we do note that the behavioral cloning accuracy gap
> between using AE/VAE/TT encodings and using full frames is small for most of the games
> we explored, so using other encoding methods would likely yield similar behavioral cloning
> accuracies.
>
> (response continued)

---

> ### Author Response · Authors · 2024-02-13
>
> (continued from above)
>
> > The results in Figure 4 and 5 appear quite low.
> – Could you comment on the accuracies and if they arise from specifically hard trajectories
> to predict, or if they are just symptoms of under-fitting BC policies?
> – Similarly, Figure 5 has several games at chance or worst, what would be causing this?
> – It is also hard to tell what the actual ability level of the base RL agents are. How do the
> raw scores obtained on each game relate to SOTA agents on Atari? Are they human-
> level? Are the trajectories representing interesting and complex behaviors?
> – Figure 4 and 5 both seem to show that using raw RGB (“no encoding” baseline) are
> the best options. This usually isn’t the case when there is some transfer gap, could you
> comment on that?
>
> As you point out, some of the accuracies and gameplay scores are low. We believe this is due
> to the amount of training data that we provided to behavioral cloning agents. In Figure 6, we
> increased the amount of training data from 40 to 160 trajectories for Seaquest and found that
> prediction accuracy increased from 47% to 55%. Given more data, we expect that gameplay
> accuracies would continue to increase. Other works in behavioral cloning have also found
> that adding more training data does tend to improve behavioral cloning performance (e.g.,
> Figure 9 in [8], Figure 3 in [2], Figure 5 in [10]). Low gameplay scores also likely stem from
> issues with distribution shift; the distribution of data used to train the agent differs from the
> distribution of data encountered while testing the agent, since the agent’s actions directly
> affect future states it encounters [9]. The low scores achieved by the behavioral cloning agents
> are also consistent with [5], which found that behavioral cloned agents from human data were
> not able to achieve human-level gameplay scores.
>
> To add some additional detail on the ability level of the base RL agents, we added a table to
> the Appendix containing a comparison of the raw scores achieved by the RL agents compared
> to random actions and to human player scores from [7] (see Table 8). In all cases the RL
> agent performs better than the randomly acting agent. However, the RL agent only achieves
> human level in two of the seven games. Although the RL agents we used to behavioral clone
> are not state-of-the-art, the RL agents still exhibit learnt behavior, as exhibited by higher
> gameplay scores than random actions. We also include the length of gameplay for the RL
> agent compared to the human player in Table 9 to show that the RL agent has learned some
> gameplay ability. In future work, we hope to use human gameplay data instead of data
> generated via a RL agent.
>
> With regards to the transfer gap, it is true that self-supervised learning is able to learn features
> which produce better downstream results than even using the original data (e.g., [3] for image
> classification and [6] for reinforcement learning). However, it seems that these approaches
> do not consistently produce better results for imitation learning [4]. Although using these
> encodings does not lead to improved behavioral cloning accuracies, one advantage of using
> the encodings was the computational speedup. It would be useful to look into this in future
> work.
>
> > The setup assumes frame-level encoding, followed by an action predictor which just consumes
> 4 frames to predict the current action.
> – Did you explore a larger number of frame stacking or other architectures for the action
> predictor?
> – How would one integrate the two and leverage videos in the representation?
>
> We chose to stack four frames, following the method used in [7]. This was not a parameter
> that we adjusted in our experiments. In terms of using more of the video data, one approach that we have not explored is compressing different segments of gameplay data, instead of just
> single frames. Given the amount of overlap / similarity between consecutive frames, such an
> approach could allow for additional compression compared to compressing individual frames.
> However, this is not something we have tested.
>
> (continued below)

---

> > ### Author Response · Authors · 2024-02-13
> >
> > (continued from above)
> >
> > > Most of my worries about the paper are really about the TT format and its implications. Given
> > this isn’t as relevant to this particular paper given its claims, I am not expecting these to be
> > addressed, but would be interested in hearing the authors’ thoughts on them:
> > – Isn’t a multilinear decomposition too “weak” to represent complex visual scenes in a way
> > that supports generalization? I would have expected non-linearities to help, compared to
> > “just” doing SVD?
> > – Does the choice of dimensions (i.e. the reshaping done on the 4-dims videos to get to
> > the 6-dim matrix) impact the performance/accuracy of the TT approximation or what it
> > focuses on?
> > – You mention that TT-SVD is limited to 30’000 frames. Does TT-ICE* have some kind
> > of limitations (e.g. is the accuracy on earlier datapoints reduced as N increases?). Would
> > it scale to O(1M) frames (imagine using the MineCLIP dataset)?
> > – How would you extend it to handle videos in a way that shares information across the
> > time axis? As far I understand, G6 is independent across frames?
> >
> > This work demonstrates that tensor decompositions are in fact capable of extracting lower
> > dimensional encodings from high dimensional image data. Furthermore, performing TT-
> > decomposition is not equivalent to just performing a single SVD on images since SVD only
> > computes the span for just one column/row space. TT-decomposition discovers additional
> > features that interact across dimensions in a separable, multi-linear fashion. The results we
> > provide in the experiments section also support this fact and demonstrate that TT results in
> > sufficiently expressive (nonlinear) representations for this case.
> >
> > The reshaping process can be seen as a hyperparameter of the decomposition. As is noted
> > in Section 3.1.3, reshaping images to have smaller dimensions is done to limit the maximum
> > TT-rank for each dimension. This also results in a slower growth in overall TT-ranks and
> > yields higher compression. However, this poses a trade-off between execution time and TT-
> > ranks. If each dimension is reshaped all the way to their prime factors, the sequential nature
> > of Algorithm 1 slows down the compression process.
> >
> > The limit of TT-SVD mainly comes from the necessity of loading all frames to be compressed
> > onto the memory at once. This problem exacerbates when an SVD is attempted on all
> > images. This increases the required memory for the computation greatly and therefore poses
> > a bottleneck. Since TT-ICE is an incremental algorithm, training images can be loaded onto
> > memory in smaller batches (in our case as batches of 2500 images), and therefore 1) reduces
> > the memory required to load the data onto memory, and 2) reduces the memory required to
> > compute SVD successfully.
> >
> > The response given to reviewer k7jk on “memory use and scalability” is in line with this
> > question as well. *”Generally, the number of parameters in the encoders scales as $O(dnr^2)$,
> > where $d$ is the number of features, $n$ is the size of each feature, and $r$ is the rank. If a
> > common set of features can be identified across all games such that $d$ is fixed, then the number of parameters will scale quadratically with rank.”* There’s no need to store encoded training
> > images as they can be re-encoded during inference time. Finally, TT-ICE explores all possible
> > bases for a given dataset without forgetting. So theoretically, the number of updates to the
> > TT-cores will decrease with increased data and stop in the limit. Please refer to Theorems 2
> > and 4 in [1] for mathematical proofs of approximation.
> >
> > The suggestion for sharing information across the time axis is very interesting to think about.
> > The current way that information is propagated is through 4-frame history. The implicit
> > assumption is that a 4-frame window captures the state of the system and is sufficient to
> > predict any future or past state (the information structure is markovian). As the question
> > alludes to, it is not guaranteed that a 4-frame history really does represent the state, and
> > indeed information about the past and future may be lost. In other dynamical system contexts
> > this is sometimes addressed by adding additional features to the state to ensure Markovianity
> > – so that each data increment captures information of the full state. If such features were
> > known, it may be possible to create extra dimensions in the tensor network representation.
> > Such features could be e.g., some time-delayed frames (skip connections over time periods),
> > or maybe handcrafted features. This is certainly an interesting question to pursue in future
> > work.

---

> > > ### Author Response · Authors · 2024-02-13
> > > **References**
> > >
> > > [1] Doruk Aksoy, David J Gorsich, Shravan Veerapaneni, and Alex A Gorodetsky. An incremental
> > > tensor train decomposition algorithm. arXiv preprint arXiv:2211.12487, 2022.
> > >
> > > [2] Brian Chen, Siddhant Tandon, David Gorsich, Alex Gorodetsky, and Shravan Veerapaneni.
> > > Behavioral cloning in atari games using a combined variational autoencoder and predictor
> > > model. In 2021 IEEE Congress on Evolutionary Computation (CEC), pages 2077–2084. IEEE,
> > > 2021.
> > >
> > > [3] Ting Chen, Simon Kornblith, Mohammad Norouzi, and Geoffrey Hinton. A simple frame-
> > > work for contrastive learning of visual representations. In Hal Daum ́e III and Aarti Singh,
> > > editors, Proceedings of the 37th International Conference on Machine Learning, volume 119 of
> > > Proceedings of Machine Learning Research, pages 1597–1607. PMLR, 13–18 Jul 2020.
> > >
> > > [4] Xin Chen, Sam Toyer, Cody Wild, Scott Emmons, Ian Fischer, Kuang-Huei Lee, Neel Alex,
> > > Steven H Wang, Ping Luo, Stuart Russell, et al. An empirical investigation of representation
> > > learning for imitation. In Thirty-fifth Conference on Neural Information Processing Systems
> > > Datasets and Benchmarks Track (Round 2), 2021.
> > >
> > > [5] Anssi Kanervisto, Joonas Pussinen, and Ville Hautam ̈aki. Benchmarking end-to-end be-
> > > havioural cloning on video games. In 2020 IEEE conference on games (CoG), pages 558–565.
> > > IEEE, 2020.
> > >
> > > [6] Kuang-Huei Lee, Ian Fischer, Anthony Z. Liu, Yijie Guo, Honglak Lee, John Canny, and
> > > Sergio Guadarrama. Predictive information accelerates learning in rl. In Proceedings of the
> > > 34th International Conference on Neural Information Processing Systems, NIPS’20, Red Hook,
> > > NY, USA, 2020. Curran Associates Inc.
> > >
> > > [7] Volodymyr Mnih, Koray Kavukcuoglu, David Silver, Andrei A. Rusu, Joel Veness, Marc G.
> > > Bellemare, Alex Graves, Martin Riedmiller, Andreas K. Fidjeland, Georg Ostrovski, Stig Pe-
> > > tersen, Charles Beattie, Amir Sadik, Ioannis Antonoglou, Helen King, Dharshan Kumaran,
> > > Daan Wierstra, Shane Legg, and Demis Hassabis. Human-level control through deep reinforce-
> > > ment learning. Nature, 518:529–533, 2015.
> > >
> > > [8] Tim Pearce and Jun Zhu. Counter-strike deathmatch with large-scale behavioural cloning. In
> > > 2022 IEEE Conference on Games (CoG), pages 104–111. IEEE, 2022.
> > >
> > > [9] St ́ephane Ross and Drew Bagnell. Efficient reductions for imitation learning. In Proceedings of
> > > the thirteenth international conference on artificial intelligence and statistics, pages 661–668.
> > > JMLR Workshop and Conference Proceedings, 2010.
> > >
> > > [10] Faraz Torabi, Garrett Warnell, and Peter Stone. Behavioral cloning from observation. In
> > > Proceedings of the Twenty-Seventh International Joint Conference on Artificial Intelligence,
> > > IJCAI-18, pages 4950–4957. International Joint Conferences on Artificial Intelligence Organi-
> > > zation, 7 2018.

---

### Decision · Action_Editor_S1KK · 2024-03-06

**Recommendation:** Accept with minor revision

**Comment:**

This paper proposed a tensor-network latent-space encoding approach to improve video-to-action game playing based on behavioral cloning.

In the initial round, the reviewers' comments are concentrated on the clarity of the presentations and the lack of broad comparisons to existing methods. The authors' rebuttal has addressed most of the reviewers' concerns. In the final recommendations, the remaining concerns are
- [Not an issue] lacking technical novelty in Tensor-Decomposition: Given the review criteria of TMLR and the application of video-to-action game playing with tensor network is new, I don't consider it as a weakness.
- [Major] inference cost: two reviewers demanded more analysis on the inference cost, in addition to the training cost.
- [Minor] comparison to other representation learning methods: to demonstrate the effectiveness of the proposed embedding method, the authors should add comparisons to other representation learning methods, such as contrastive learning. Though the reviewers also acknowledged that the current results already show performance advantages.

Based on my own reading and the reviewers' suggestions, I recommend accept with minor revision. The authors should at least address the major issue (second bullet point) in the revised version.

**Audience:**

This paper should be of broad interest, given that the topic intersects deep learning and reinforcement learning.

**Claims And Evidence:**

Yes. While the empirical comparisons can be further expanded to include more baselines, all reviewers validated the claims of the performance improvements for Low-Rank Tensor-Network Encodings of the considered tasks.